# Multiplex precise base editing in cynomolgus monkeys

Wenhui Zhang[1,2,7], Tomomi Aida[3,7,8✉], Ricardo C. H. del Rosario [4,7], Jonathan J. Wilde [3,7], Chenhui Ding[5], Xiaohui Zhang[6], Zulqurain Baloch [1], Yan Huang[1], Yu Tang[1], Duanduan Li[1], Hongyu Lu[1], Yang Zhou[3], Minqing Jiang[3], Dongdong Xu[1], Zhihao Fang[1], Zhanhong Zheng[1,2], Qunshan Huang[1], Guoping Feng [3,4,8✉] & Shihua Yang[1,2,8✉]

Common polygenic diseases result from compounded risk contributed by multiple genetic variants, meaning that simultaneous correction or introduction of single nucleotide variants is required for disease modeling and gene therapy. Here, we show precise, efficient, and simultaneous multiplex base editing of up to three target sites across 11 genes/loci in cynomolgus monkey embryos using CRISPR-based cytidine- and adenine-base editors. Unbiased whole genome sequencing demonstrates high specificity of base editing in monkey embryos. Our data demonstrate feasibility of multiplex base editing for polygenic disease modeling in primate zygotes.

[1] College of Veterinary Medicine, South China Agricultural University, 510642 Guangzhou, China. [2] Guangdong Laboratory of Lingnan Modern Agriculture, 510642 Guangzhou, China. [3] McGovern Institute for Brain Research, Department of Brain and Cognitive Sciences, Massachusetts Institute of Technology, Cambridge, MA 02139, USA. [4] Stanley Center for Psychiatric Research, Broad Institute of MIT and Harvard, Cambridge, MA 02142, USA. [5] Key Laboratory of Reproductive Medicine of Guangdong Province, Reproductive Medicine Center, the First Affiliated Hospital of Sun Yat-sen University, Guangzhou 510080, China. [6] Shanghai Key Laboratory of Regulatory Biology, Institute of Biomedical Sciences and School of Life Sciences, East China Normal University, Shanghai 200241, China. [7]These authors contributed equally: Wenhui Zhang, Tomomi Aida, Ricardo C.H. del Rosario, Jonathan J. Wilde. [8]These authors jointly supervised this work: Tomomi Aida, Guoping Feng, Shihua Yang. ✉email: aidat@mit.edu; fengg@mit.edu; yangsh@scau.edu.cn

Genome-wide association studies have identified thousands of single nucleotide variations (SNVs) associated with increased risk for various common human diseases. However, because each SNV only contributes a small amount to disease risk, common diseases are thought to be caused by accumulated risk from multiple SNVs[1]. To model complex common diseases in animals, thus, it is necessary to develop a method for simultaneous and precise installation of multiple SNVs by genome editing. The CRISPR (clustered regularly interspaced short palindromic repeats)/Cas9 (CRISPR-associated protein 9) system has enabled single nucleotide corrections in both somatic cells and zygotes in various species including humans[2,3]. However, it is still difficult to control the output of the editing because DNA double-strand breaks (DSBs) induced by CRISPR are primarily repaired through non-homologous end-joining (NHEJ), which can lead to the formation of small insertions or deletions (indels), rather than precise base correction by homologous recombination (HR) in the presence of a donor DNA template. As such, low HR frequency makes multiplexed editing very difficult. As an alternative to DSB-mediated HR for gene editing, targeted base editing by cytidine deaminase or adenosine deaminase Cas9 fusion proteins (using Cas9 nickase or catalytically-dead Cas9), which convert cytosine (C) to thymine (T) or adenine (A) to guanine (G) without DSB induction, has recently been developed[4–6]. Thus, base editing has a great potential for precise therapeutic gene correction and disease modeling[7–10].

Cynomolgus monkey (Macaca fascicularis) is the animal model best suited for biomedical research that requires a model organism with a close relationship to humans in both genetic and physiological aspects[11,12]. Thus, targeted precise genome editing in cynomolgus monkey is critically important not only for human disease modeling, but also for the development of gene therapies. However, the long cynomolgus reproduction cycle (5–6 years)[13] makes it unfeasible to produce polygenic monkey models carrying multiple SNVs by crossing different single SNV mutant monkeys (>10 years would be required). To overcome this limitation, efficient multiplex genome editing is required to generate disease models in F0 animals. Although CRISPR-Cas9 has been shown to efficiently work in cynomolgus monkey zygotes for knockout monkey production[12,14,15], it is still unknown whether base editing works in cynomolgus monkey zygotes for targeted base conversions. Here, using cynomolgus monkey zygotes, we demonstrate the feasibility of precise, efficient, and multiplexed base editing and identify editing strategies best suited for the production of F0 animals with polygenic traits.

## Results

**Single C-to-T base editing in monkey embryos.** First, we tested whether targeted base editing works in cynomolgus monkey embryos using BE3, a cytosine base editor (CBE) consisting of the Cas9 nickase fused to rAPOBEC1 and a uracil glycosylase inhibitor[5]. We chose the gene fumarylacetoacetate hydrolase (FAH), in which mutations cause tyrosinemia[16], as an initial target for our studies. Currently, ~100 mutations have been found in hereditary tyrosinemia type 1 (HT1) patients[16]. We designed a single guide RNA (sgRNA) to introduce FAH W78X, which results in a stop codon in exon 4 of FAH (Fig. 1a) and was reported as a causative mutation in HT1 patients[17]. We prepared sgRNA and BE3 mRNA and injected them into the cytoplasm of cynomolgus monkey zygotes (Fig. 1b). Injected embryos were then cultured for 3–4 days and collected for genotyping. To genotype each embryo, whole-genome amplification (WGA) was performed and the target site was amplified by PCR, cloned, and analyzed by Sanger sequencing. These genotyping procedures did not introduce any mutations at the target sites in uninjected wild-type embryos (Supplementary Table 1). In exon 4 of FAH, two Cs (C5 and C6) are located within the BE3 activity window of the FAH sgRNA (core positions 4–8) and the conversion of the two Cs to Ts introduces the W78X mutation (Fig. 1a). We found that 11 out of 16 injected embryos had genomic modifications at the target site in the FAH exon 4 locus (Table 1). Two embryos (#7 and #8) had targeted C-to-T (G-to-A) conversions with 100% allele frequency at both C5 and C6 within sequenced clones (Fig. 1c, h, Supplementary Fig. 1a). An additional eight embryos had targeted C-to-T (G-to-A) conversions with allele frequencies that ranged between 12.1% and 100% for at least one C (Fig. 1c, Supplementary Fig. 1a). These embryos were mosaic with wild-type and/or non-C-to-T conversions at each C (Fig. 1c, e, Supplementary Fig. 1a). Indels were also found in 3 out of 11 edited embryos (Fig. 1c, g, Supplementary Fig. 1a). One embryo (#10) had C-to-A (G-to-T) conversion with 100% allele frequency only at C5 (Fig. 1c, e, Supplementary Fig. 1a). As a result, 10 out of 11 edited embryos had W78X stop mutations with allele frequencies ranging between 12.1% and 100% by targeted C-to-T conversion (Supplementary Fig. 1a). Because BE3 often induces non-C-to-T conversions and indels (Fig. 1c, e, g, Table 1, Supplementary Fig. 1a), a BE variant with increased accuracy, BE4-Gam, was tested using the same FAH sgRNA as used with BE3 (ref. [18]). Although BE4-Gam only induced C-to-T conversions, its conversion frequency was much lower than that of BE3 (Supplementary Fig. 2a–c). Furthermore, BE4-Gam induced indels at similar rates as BE3 (Supplementary Fig. 2a, d and Supplementary Table 2). Thus, we continued to use BE3 for the rest of the experiments in this study due to the requirement of high editing efficiency in cynomolgus.

**Single A-to-G base editing in monkey embryos.** Next, we tested A-to-G base editing in cynomolgus monkey embryos using adenine base editor (ABE), which consists of the Cas9 nickase fused with TadA[4]. As an initial target gene for A-to-G base editing, we chose amyloid precursor protein (APP), which is mutated in familial early-onset Alzheimer's disease[19]. We designed an sgRNA to introduce two missense mutations into exon 2 of APP (Fig. 1a) and injected the sgRNA together with ABE7.10 mRNA (Fig. 1b). Proper A-to-G editing of the two As (A5 and A7) within the ABE7.10 activity window introduces the Q33R and I34V missense mutations into APP (Fig. 1a). We found that 6 out of 9 injected embryos had genomic modifications at the target site in the APP exon 2 locus (Table 1). These were all A-to-G conversions with allele frequencies that ranged between 20% and 88.6% for at least at one A (Fig. 1d, i, Supplementary Fig. 1b). A7 was converted to G more than A5 (six out of six embryos for A7 and three out of six embryos for A5). All of the modified embryos were incompletely edited and mosaic with the wild-type allele (Fig. 1d, Supplementary Fig. 1b). Importantly, neither non-A-to-G conversions nor indels were detected (Fig. 1d, f, g, Supplementary Fig. 1b). As a result, six out of six edited embryos had the I34V mutation and three out of six edited embryos had both Q33R and I34V mutations with allele frequencies that ranged between 20% and 75% by targeted A-to-G conversion (Supplementary Fig. 1b). These results suggest that both BE3- and ABE-mediated targeted base editing are effective and precise in cynomolgus monkey embryos.

**Multiplex C-to-T or A-to-G base editing in monkey embryos.** Encouraged by these results, we next tried to simultaneously edit multiple loci by co-injection of sgRNAs targeting multiple sites with BE3 or ABE in cynomolgus monkey embryos. To this end, we first designed multiple sgRNAs for double and triple editing of

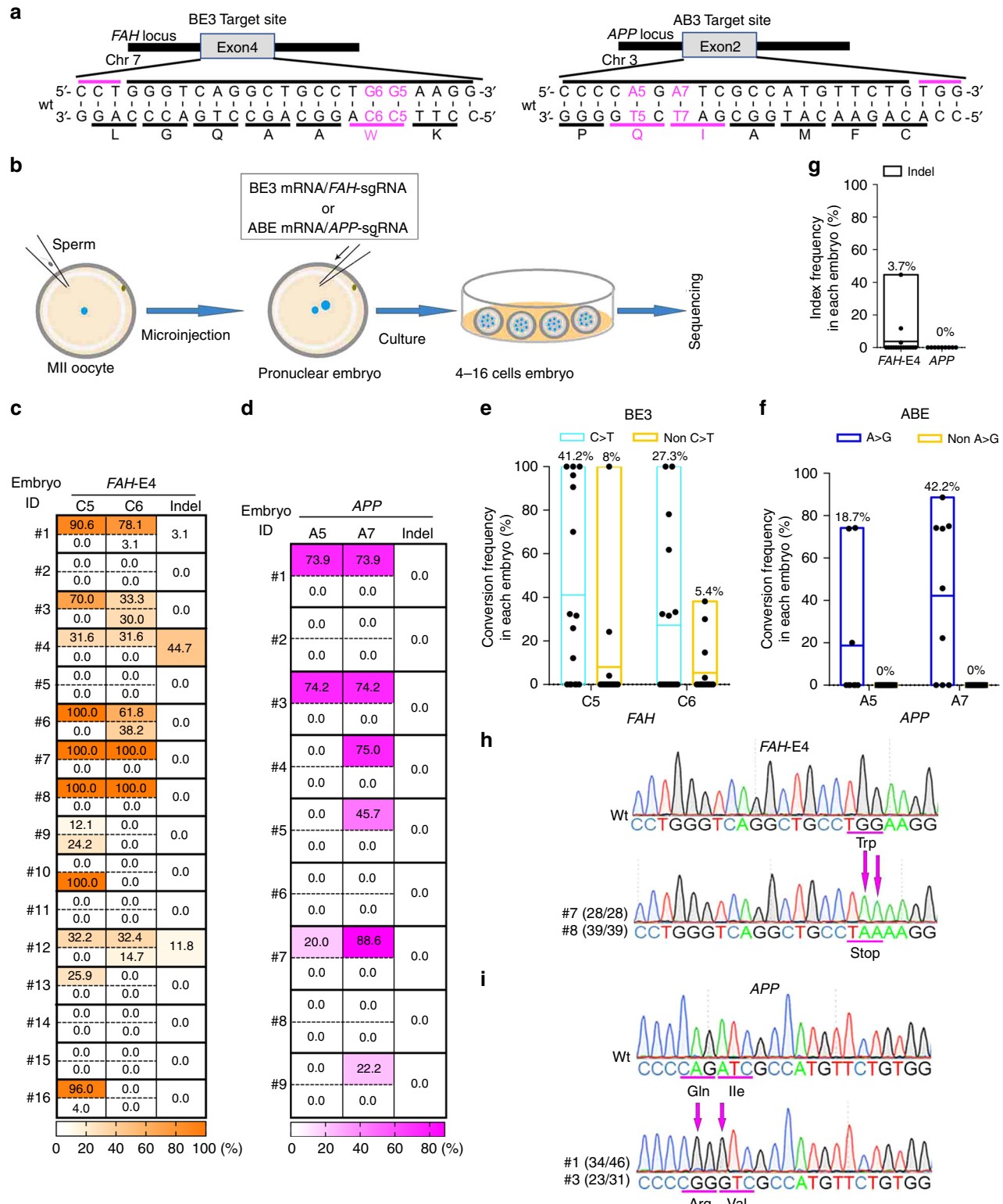

**Fig. 1 Single C-to-T or A-to-G base editing in monkey embryos. a** Target sites of *FAH* exon 4 for C-to-T base editing by BE3 and *APP* for A-to-G base editing by ABE. Top black bars: sgRNA spacer sequences, top magenta bars: PAM sequences, bottom black bars: amino acid codons, bottom magenta bars: target amino acids to be base edited. Wt: wild-type. **b** Schematic representation of base editing in monkey zygotes. **c, d** Base editing outcomes by (**c**) BE3 and (**d**) ABE. Numbers shown on the left of the tables indicate embryo ID. In each embryo, top and bottom rows indicate intended (C-to-T or A-to-G) and unintended (non-C-to-T or non-A-to-G) conversions, respectively. **e, f** Efficiencies and accuracies of base editing outcomes by (**e**) BE3 and (**f**) ABE. **g** Indel frequencies. In **e–g** data are represented as boxplots where the middle line is the mean (also shown as value on top of the bar) and the top and bottom lines correspond to the maximum and minimum mutant allele frequencies. Each dot represents one embryo. *FAH*-E4: injections were performed thrice (total *n* = 16); *APP*: injections were performed twice (total *n* = 9). **h, i** representative sequence chromatograms of cloned PCR amplicons. Numbers of base conversion-positive clones and the total clones sequenced are shown in parentheses.

**Table 1 Single C-to-T or A-to-G base editing in monkey embryos.**

| Editor | Target sites | Analyzed | Edited | Indel | C > T or A > G | 100% C > T or A > G |
|---|---|---|---|---|---|---|
| BE3 | *FAH* | 16 | 11 (68.8%) | 3 (27.3%) | 10 (91%) | 2 (20%) |
| ABE | *APP* | 9 | 6 (66.7%) | 0 (0%) | 6 (100%) | 0 (0%) |

*FAH*-E4 injections were performed thrice, *APP* injections were performed twice.

*FAH* targets with BE3 (Fig. 2a). Then, the BE3 mRNA and two or three sgRNAs were co-injected. For double base editing at *FAH* exons 7 and 9, seven Cs (C4-5 and C7 in exon 7, and C4 and C6-8 in exon 9 sgRNA1) locate within the BE3 activity window of *FAH* exon 7 and 9 sgRNAs (core positions 4–8) (Fig. 2a). We found that 8 out of 9 injected embryos had genomic modifications at target sites in the *FAH* exon 7 or 9 locus (Table 2). Importantly, 5 embryos had targeted C-to-T (G-to-A) conversions at both sites with allele frequencies that ranged between 1.7% and 100% for at least one C. C7 in exon 7 was efficiently converted to T (100% in 3 embryos), but the conversions of other Cs were incomplete (Fig. 2b, Supplementary Fig. 3a). All embryos were mosaic with wild-type and/or non C-to-T conversions at each C (Fig. 2b, e, Supplementary Fig. 3a) and indels were also observed (Fig. 2b, f, Supplementary Fig. 3a). Interestingly, only C7 in exon 7 and C4 and C6 in exon 9 were efficiently converted, whereas the other flanking Cs were not converted or had fewer conversions (Fig. 2b, Supplementary Fig. 3a).

For triple base editing at *FAH* exons 4, 9, and 14, seven Cs (C5-6 in exon 4, C4, C6, and C8 in exon 9, and C6-7 in exon 14) locate within the BE3 activity window (core positions 4–8) (Fig. 2a). We found that 8 out of 10 injected embryos had base modifications at target sites in *FAH* exon 4, 9, or 14 locus (Table 2). Two embryos (#6 and #9) had targeted C-to-T (G-to-A) conversions at all three sites with allele frequencies that ranged between 14.3% and 100% for at least one C in each site (Fig. 2c, Table 2, Supplementary Fig. 3b, e). Remarkably, one embryo (#9) had 5 C-to-T conversions across 3 exons with 100% frequencies within sequenced clones, although the other flanking Cs were not converted or had fewer conversions (Fig. 2c, Table 2, Supplementary Fig. 3b, e). The other 6 embryos had targeted C-to-T (G-to-A) conversions at 1 or 2 sites with allele frequencies that ranged between 8.3% and 100% for at least one C (Fig. 2c, Table 2, Supplementary Fig. 3b). These embryos were also incompletely edited and mosaic with wild-type and/or non-C-to-T conversions at each C (Fig. 2c, e, Table 2, Supplementary Fig. 3b). C4 in exon 9 sgRNA2 locates within the BE3 core activity window, but was not efficiently edited (only 1 in 10 embryos) (Fig. 2c, Supplementary Fig. 3b). Also, Cs located outside of the BE3 core activity window were edited in a several embryos (Fig. 2c, Supplementary Fig. 3b) and indels were also observed (Fig. 2c, f, Supplementary Fig. 3b).

Next, using ABE we designed sgRNAs to simultaneously target two genes, Hemoglobin Beta (*HBB*) and Tumor Protein p53 (*TP53*), in which mutations cause beta-thalassemia and cancer, respectively[20,21] (Fig. 2a). Three As (A4 and A5 in *HBB*, and A7 in *TP53*) locate within the ABE activity window of *HBB* and *TP53* sgRNAs (core positions 4–7) (Fig. 2a). We found that all of the 5 injected embryos only had A-to-G conversions at the target site in both the *HBB* and/or *TP53* loci (Fig. 2d, Table 2, Supplementary Fig. 3c, f). Importantly, neither non-A-to-G conversion nor indels were detected (Fig. 2d–f, Supplementary Fig. 3c, f). Four embryos had targeted A-to-G conversions at both sites with allele frequencies that ranged between 5.9% and 40.7% for at least at one A (Fig. 2d, Table 2, Supplementary Fig. 3c, f). These target sites were incompletely converted and the embryos were mosaic with wild-type at each A (Supplementary Fig. 3c).

To investigate whether multiplex base editing occurs in individual cells, we first performed long-range PCR across *FAH* exons 7-9 in BE3-double injected embryos, cloned 4 kb PCR amplicons, and sequenced the clones. C-to-T conversions in both exons 7 and 9 were detected in single clones (Supplementary Fig. 4). As a second method of examining multiplex editing in single cells, we also performed single blastomere genotyping (Supplementary Fig. 5a). In BE3 double base editing at *FAH* exons 7 and 9, 3 out of 6 embryos had double C-to-T conversions in single blastomeres (Fig. 2g–h, Supplementary Fig. 5b–d). Embryo 1 had 2 double-edited homozygous blastomeres among 4 blastomeres; embryo 3 had 3 double-edited homozygous or heterozygous blastomeres among 6 blastomeres; and embryo 5 had 1 double-edited homozygous blastomere among 4 blastomeres (Fig. 2g, Supplementary Fig. 5b, c). Overall double-editing efficiency in a single blastomere was 19.4% (6 out of 31 blastomeres in 6 embryos) (Fig. 2h). Other blastomeres were single-edited with C-to-T conversion (19.4% at *FAH* exons 7 and 6.5% at *FAH* exon 9), indels (6.5%), and non-edited wild-type alleles (Fig. 2h). In BE3 triple base editing at *FAH* exons 4, 9, and 14, 2 out of 6 embryos had triple C-to-T conversions in single blastomeres (Fig. 2g–h, Supplementary Fig. 5b–d). Embryo 2 had 4 triple-edited homozygous or heterozygous blastomeres among 5 blastomeres; and embryo 5 had 2 triple-edited heterozygous blastomere among 4 blastomeres (Fig. 2g, Supplementary Fig. 5b, c). Overall, triple editing efficiency in a single blastomere was 22.2% (6 out of 27 blastomeres in 6 embryos) (Fig. 2h). Other blastomeres were single (18.5% at *FAH* exons 4) or double (7.4% at *FAH* exons 9 and 14) edited with C-to-T conversion, or edited with indel or non-C-to-T conversions (14.8%), and non-edited wild-type sequence (Fig. 2h).

In ABE double base editing at *HBB* and *TP53*, 2 out of 3 embryos had double A-to-G conversions in single blastomeres (Fig. 2g–h). Embryo 1 had 2 double-edited homozygous or heterozygous blastomeres among 4 blastomeres; and embryo 2 had 3 double-edited homozygous or heterozygous blastomeres among 5 blastomeres (Fig. 2g, Supplementary Fig. 5b, c). Overall double editing efficiency in a single blastomere was 38.5% (5 out of 13 blastomeres in 3 embryos) (Fig. 2h). Other blastomeres were single edited with A-to-G conversion (7.7% at *HBB* and 23.1% at *TP53*), and non-edited wild-type allele (Fig. 2h). These results indicate the feasibility of BE3- or ABE-mediated multiple targeted base editing at up to three target genes/sites in a single blastomere of cynomolgus monkey embryos at the same time, but also highlight the necessity of further improvements of efficiency and accuracy of base editing.

**Multiplex C-to-T and A-to-G base editing in monkey embryos.**
Finally, we further tried to improve the accuracy and efficiency of multiplex base editing, especially CBE, by combining Streptococcus pyogenes Cas9 (SpCas9)-based ABE and Staphylococcus aureus Cas9 (SaCas9)-based SaKKH-BE3 (ref. [22]) to avoid sgRNA crosstalk between ABE and BE3. For triple C-to-T and G-to-A base editing, we designed two SaKKH-BE3-sgRNAs for Empty Spiracles Homeobox 1 (*EMX1*) and FA complementation group F (*FANCF*) and an SpABE-sgRNA for *BRCA1*. The three genes are associated with Kallmann Syndrome[23], Fanconi anemia[24], and

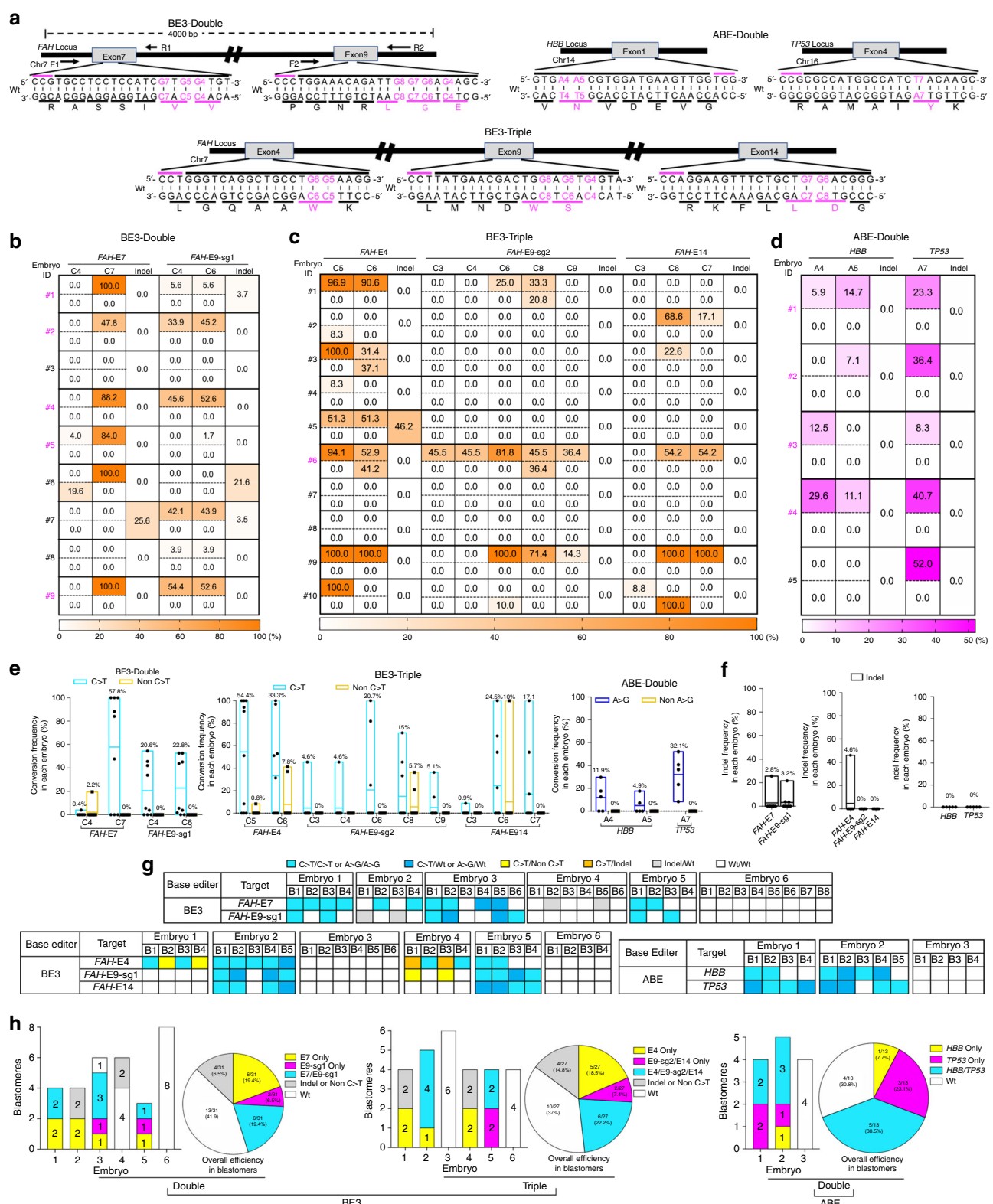

cancers[25], respectively (Fig. 3a). Six Cs (C7-8 in *EMX1* and C4 and C6-8 and C12 in *FANCF*) locate within SaKKH-BE3 activity window of *EMX1* and *FANCF* sgRNAs (core positions 3-12) for double C-to-T base editing and an A5 in *BRCA1* locates within the ABE activity window of the *BRCA1* sgRNA (core positions 4–7) (Fig. 3a). SaKKH-BE3 and SpABE mRNAs, along with the three sgRNAs were co-injected and embryos were genotyped following the previously described methods. Surprisingly, we

found that all 8 of the injected embryos had genomic modifications at target sites in *EMX1* and *FANCF* and 5 out of 8 embryos at a target site in *BRCA1* (Fig. 3b, Table 3, Supplementary Fig. 6). All embryos had SaKKH-BE3-mediated C-to-T conversions at target sites in *EMX1* and *FANCF* (Fig. 3b, Table 3, Supplementary Fig. 6) and, remarkably, the C-to-T conversion frequencies by ranged from 47% to 100%, with 6 out of 8 embryos showing greater than 60% conversion across all 6 Cs. Additionally, all 6 Cs

**Fig. 2 Double/triple multiplex C-to-T or A-to-G base editing in monkey embryos. a** Target sites of *FAH* exons 4, 7, 9 and 14 for double/triple C-to-T base editing by BE3, and *HBB* and *TP53* for double A-to-G base editing by ABE. Top black bars: sgRNA spacer sequences, top magenta bars: PAM sequences, bottom black bars: amino acid codons, bottom magenta bars: target amino acids to be base edited. Wt: wild-type. **b–d** Base editing outcomes. **b** Double or **c** triple multiplex C-to-T base editing by BE3 and **d** double multiplex A-to-G base editing by ABE are shown. Numbers shown in the left of the tables indicate embryo ID. Magenta indicates **b, d** double and **c** triple base-edited embryos. In each embryo, top and bottom rows indicate intended (C-to-T or A-to-G) and unintended (non-C-to-T or non-A-to-G) conversions, respectively. **e** Efficiencies and accuracies of base editing outcomes. **f** Indel frequencies. In **e–f** data are represented as boxplots where the middle line is the mean (also shown as value on top of the bar) and the top and bottom lines correspond to the maximum and minimum mutant allele frequencies. Each dot represents one embryo. BE3-Double: injections were performed twice (total $n = 9$); BE3-Triple: injections were performed twice (total $n = 10$); ABE-Double: injections were performed twice (total $n = 5$). **g, h** Single blastomere genotyping. **g** Single blastomere genotyping results in each blastomere are shown by color codes. B blastomere. **h** Single blastomere genotyping results for each embryo (bar graphs) or blastomeres from all embryos (pie charts).

**Table 2 Double/triple multiplex C-to-T or A-to-G base editing in monkey embryos.**

| Editor | # of target | Target sites | Analyzed | Edited | Single | Double | Triple | 100% Single | 100% Double | 100% Triple |
|---|---|---|---|---|---|---|---|---|---|---|
| BE3 | 2 | *FAH*-E7/E9-1 | 9 | 8 (88.9%) | 3 (37.5%) | 5 (62.5%) | – | 2 (66.7%) | 0 (0%) | – |
| BE3 | 3 | *FAH*-E4/E9-2/E14 | 10 | 8 (80%) | 3 (37.5%) | 3 (37.5%) | 2 (25%) | 0 (0%) | 0 (0%) | 1 (50%) |
| ABE | 2 | *HBB/TP53* | 5 | 5 (100%) | 1 (20%) | 4 (80%) | – | 0 (0%) | 0 (0%) | – |

Single, double, or triple represents single, double, or triple C > T or A > G conversions. 100% Single, 100% Double, or 100% Triple represents 100% single, 100% double, or 100% triple C > T or A > G conversions. Injections were performed twice.

within the target window were converted in all embryos (Fig. 3b, Supplementary Fig. 6). Importantly, no indels were detected and we only observed non-C-to-T conversion at a low level (18%) at a single base in one embryo (Fig. 3c–e, Supplementary Fig. 6). At the *BRCA1* SpABE target site, five out of eight embryos showed some degree of A-to-G conversion (Fig. 3b, Table 3, Supplementary Fig. 6). The A-to-G conversion frequency at *BRCA1* in individual embryos ranged from 10% to 86% (Fig. 3b, Supplementary Fig. 6) and neither non-A-to-G conversion nor indels were detected (Fig. 3b, d, e, Supplementary Fig. 6). Importantly, the 5 embryos with A-to-G conversions at the target site of *BRCA1* also had high levels of C-to-T conversion at target sites in *EMX1* and *FANCF*, demonstrating feasible and efficient multiplex C-to-T and A-to-G base editing (Fig. 3b, Table 3, Supplementary Fig. 6).

Next, we performed single blastomere genotyping to investigate whether multiplex base editing occurred in single cells. With respect to SaKKH-BE3 double base editing at *EMX1* and *FANCF*, all embryos had double C-to-T conversions in single blastomeres (Fig. 3f–g, Supplementary Fig. 7). Furthermore, all blastomeres except for 2 blastomeres in embryo 6 had double homozygous (29/37) or heterozygous (8/37) C-to-T conversions (Fig. 3f, Supplementary Fig. 7). Remarkably, embryo 7 was homozygous non-mosaic embryo (Fig. 3f, Supplementary Fig. 7). In line with our results from bulk genotyped embryos, no blastomeres were edited with non-C-to-T conversions or indels (Supplementary Fig. 7). Overall, double C-to-T editing efficiency in a single blastomere reached 94.9% (37 out of 39 blastomeres in 7 embryos) (Fig. 3g) with all other blastomeres genotyping as unedited wild-type (Fig. 3f–g, Supplementary Fig. 7). At the *BRCA1* ABE locus, three out of seven embryos had A-to-G conversions in single blastomeres (Fig. 3f–g, Supplementary Fig. 7). Embryo 1 had 3 homozygous or heterozygous edited blastomeres among 6 blastomeres; embryo 3 had 2 heterozygous edited blastomeres among four blastomeres; and embryo 5 had 4 homozygous or heterozygous edited blastomeres among six blastomeres (Fig. 3f, Supplementary Fig. 7). Total A-to-G editing efficiency in a single blastomere was 23.1% (9 out of 39 blastomeres in 7 embryos) (Fig. 3g) with all other blastomeres genotyping as wild-type (Fig. 3f–g, Supplementary Fig. 7).

Importantly, 3 out of 7 embryos had triple C-to-G and A-to-G conversions in single blastomeres (Fig. 3f–g, Supplementary Fig. 7). Embryo 1 had 3 homozygous or heterozygous triple edited blastomeres among 6 blastomeres; embryo 3 had 2 heterozygous triple edited blastomeres among 4 blastomeres; and embryo 5 had 4 homozygous or heterozygous edited blastomeres among 6 blastomeres (Fig. 3f, Supplementary Fig. 7). Overall, triple C-to-T and A-to-G editing efficiency in a single blastomere was 23.1% (9 out of 39 blastomeres in 7 embryos) (Fig. 3g). No blastomeres were edited with non-C-to-T or non-A-to-G conversions or indels (Fig. 3f–g, Supplementary Fig. 7). Again, we observed mosaicism mainly due to less efficient editing of *BRCA1* by SpABE, whereas SaKKH-BE3 edited its targets with efficiency approaching 100% (Fig. 3f–g, Supplementary Fig. 7). These results indicate that SaKKH-BE3- and SpABE-mediated multi-plexed base editing works efficiently and precisely at up to three target genes in a single blastomere of cynomolgus monkey embryos.

**Off-target analyses.** To characterize the genome-wide consequences of targeted base editing in cynomolgus monkey embryos, we performed both targeted and unbiased analyses to search for off-target mutations. First, we picked the top 5 off-target candidate sites for each sgRNA as predicted by Benchling and BLAST search (Supplementary Data 1), PCR amplified them, and analyzed the loci with the T7E1 assay (Supplementary Data 2). At all the analyzed candidate sites, we found no off-target modification (Supplementary Fig. 8 and Supplementary Table 3). Next, we generated an *FAH* mutant fetus by BE3 (Fig. 4a, Supplementary Table 4 and 5). We injected sgRNA and BE3 mRNA into the cytoplasm of cynomolgus monkey zygotes as described in Fig. 1, then cultured the embryos before transfer into surrogate mothers (Supplementary Tables 4 and 5). Among 67 injected embryos with pronuclei, 56 embryos developed to between the 8-cell and blastocyst stages and were transferred into 5 surrogate mothers, resulting in 3 pregnant monkeys with 3 fetuses. We collected amniotic fluid for fetal genotyping and identified one fetus as an *FAH* W78Y mutant caused by both C-to-T and non-C-to-T conversions (Supplementary Table 5). Unfortunately, all pregnancies were miscarried after amniotic fluid collection, so we

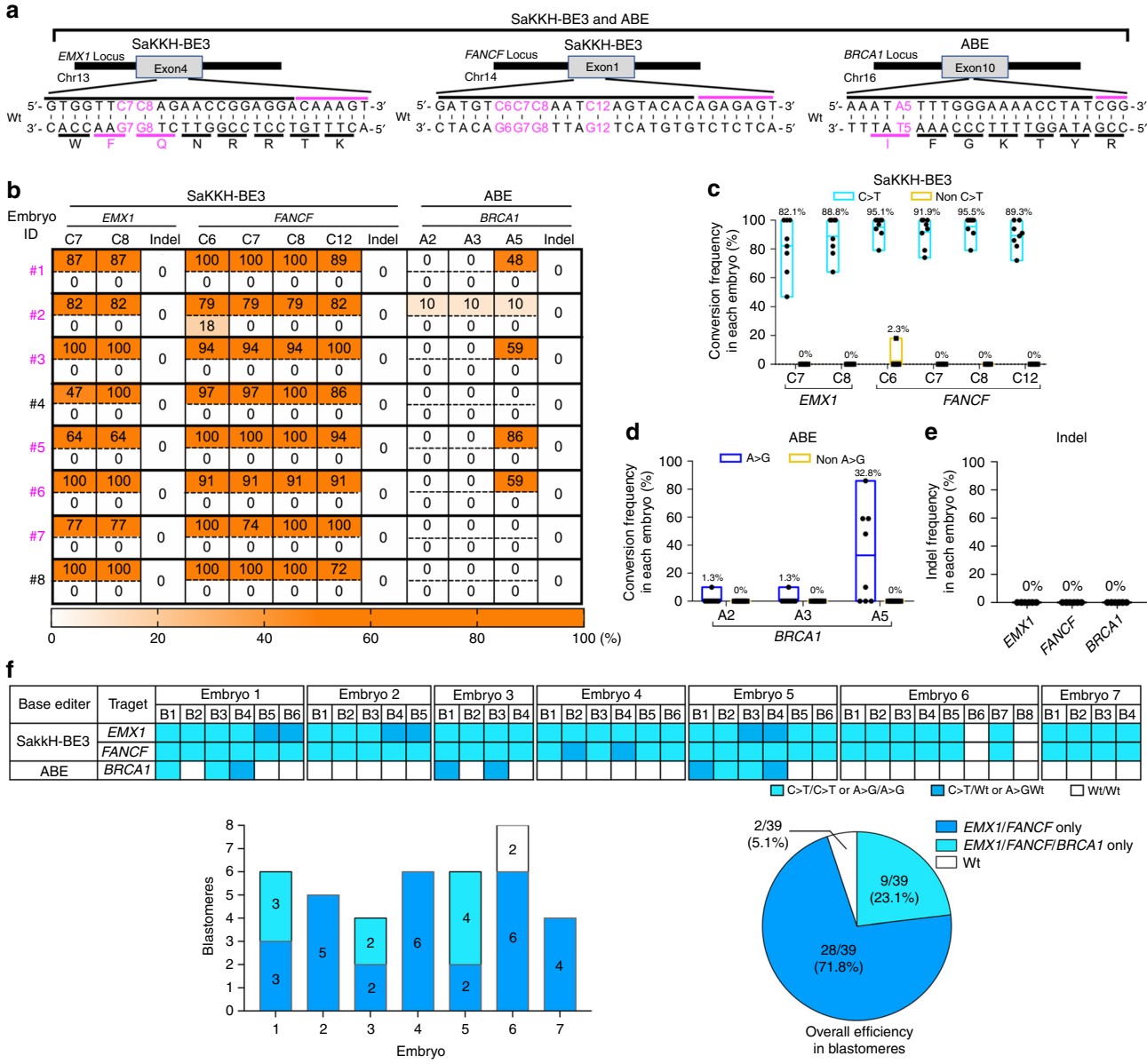

**Fig. 3 Triple multiplex C-to-T and A-to-G base editing in monkey embryos. a** Target sites of *EMX1*, *FANCF* and *BRCA1* for triple multiplex C-to-T and A-to-G base editing by SaKKH-BE3 and ABE. Top black bars: sgRNA spacer sequences, top magenta bars: PAM sequences, bottom black bars: amino acid codons, bottom magenta bars: target amino acids to be base edited. Wt: wild-type. **b** Triple C-to-T and A-to-G base editing outcomes by SaKKH-BE3 and ABE. Numbers shown to the right of the tables indicate embryo ID. Magenta indicates triple base-edited embryos. In each embryo, top and bottom rows indicate intended (C-to-T or A-to-G) and unintended (non-C-to-T or non-A-to-G) conversions, respectively. **c, d** Efficiencies and accuracies of base editing by outcomes by (**c**) SaKKH-BE3 and (**d**) ABE. Each dot represents one embryo. **e** Indel frequencies. Each dot represents one embryo. In **c–e** data are represented as boxplots where the middle line is the mean (also shown as value on top of the bar) and the top and bottom lines correspond to the maximum and minimum mutant allele frequencies. Each dot represents one embryo. Injections were performed twice (total n = 8). **f, g** Single blastomere genotyping. **f** Single blastomere genotyping results in each blastomere are shown by color codes. B blastomere. **g** Single blastomere genotyping results for each embryo (bar graphs) or blastomeres of all embryos (pie charts).

**Table 3 Triple multiplex C-to-T and A-to-G base editing in monkey embryos.**

| Editor | # of target | Target sites | | Analyzed | Edited | Single | Double | Triple | 100% Single | |
|---|---|---|---|---|---|---|---|---|---|---|
| | | **100% Double** | | | | | | | **100% Triple** | |
| SaKKH-BE3 ABE | 3 | *EMX1/FANCF BRCA1* | | 8 | | 8 (100%) | 0 (0%) | 3 (37.5%) | 5 (62.5%) | 0 (0%) |
| 2 (66.7%) | 0 (0%) | | | | | | | | | |

Single, double, or triple represents single, double, or triple C > T and/or A > G conversions. 100% Single, 100% Double, or 100% Triple represents 100% single, 100% double, and/or 100% triple C > T or A > G conversions. Injections were performed twice.

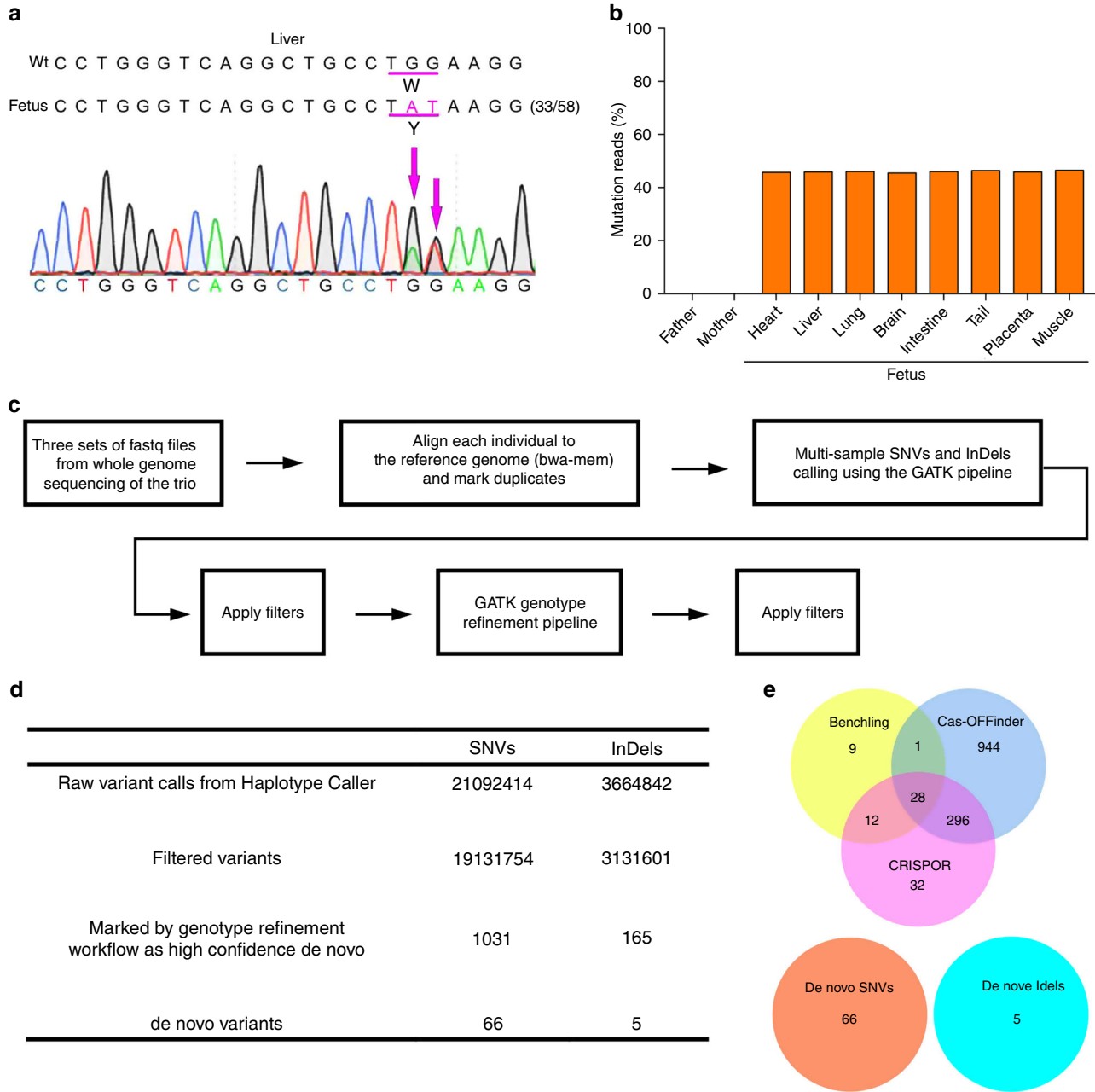

**Fig. 4 Off-target analysis. a** *FAH* on-target genotyping in liver of *FAH* mutant fetus. Bulk Sanger sequence chromatogram is shown. Target site is underlined. Numbers of base conversion-positive clones and the total clones sequenced are shown in parentheses. Wt wild-type, Fetus: *FAH* mutant fetus. **b** Deep sequencing analysis. Deep sequencing data from one father (blood), one mother (blood), and one fetus (heart, liver, lung, brain, intestine, tail, placenta, and muscle) are shown. Sequencing read numbers are as follows: Father: $n = 154,718$; mother: $n = 122,083$; heart: $n = 67,112$; liver $n = 52,976$; lung: $n = 51,608$; brain: $n = 48,480$; intestine: $n = 46,350$; tail: $n = 48,008$; placenta: $n = 69,593$ and muscle: $n = 68,145$. **c** Analytical pipeline of trio-based whole-genome sequencing. **d** Summary of trio-based whole-genome sequencing. **e** Venn diagram of in silico predicted potential *FAH*-sgRNA off-target sites and de novo mutations.

collected fetal tissue and analyzed eight organs by deep sequencing. This confirmed the *FAH* W78Y mutation in one fetus with ~50% allele frequency across organs (Fig. 4a, b). Whole-genome sequencing of the mutant fetus and its parents (30-40x Illumina paired-end sequencing) was performed, followed by analysis with the GATK SNP calling pipeline and the GATK Genotype Refinement pipeline to detect de novo variants in the mutant genome (see Methods) (Fig. 4c). From this analysis, we identified 66 de novo SNVs and 5 de novo indels (Fig. 4d, Supplementary Fig. 9, Supplementary Data 3), but none of these 71 de novo variants overlapped with any of the predicted potential off-target

sites of the *FAH* sgRNA (Fig. 4e, Supplementary Data 4–6). Two recent papers showed that the cytidine deaminase domain of BE3 can induce random C-to-T (or G-to-A) conversion throughout the genome in mice and rice in a guide RNA-independent manner[26,27]. To analyze this possibility in our results, we used the cynomolgus macaque SNPs from dbSNP/EVA to estimate the expected fraction of SNPs for each of the 9 possible base substitutions (Supplementary Table 6). We estimated the number of off-target C-to-T (G-to-A) as the observed C-to-T (G-to-A) de novo counts minus expected (total de novo SNVs times fraction of C-to-T (G-to-A) in dbSNP/EVA), and obtained 2.3 and 1.3, for

C-to-T and G-to-A, respectively. Thus, the low number of de novo SNVs and indels in our mutant, together with our low estimate of de novo BE3 off-targets (3.6 SNVs) suggest that we could not detect clear guide RNA-independent C-to-T (or G-to-A) conversion by BE3 beyond the range of germline de novo SNVs per generation in cynomolgus macaque. Taken together, these results suggest that base editing is highly specific in cynomolgus monkey embryos, although due to the limitation of the experimental design, we could not distinguish BE3-induced off-target mutation from naturally occurring de novo mutation.

## Discussion

Because it does not require the induction of double-strand breaks, base editing has great potential for precise therapeutic gene correction and disease modeling. Base editors are continuously being improved and numerous variants with enhanced efficiency, narrower target windows, and broader PAM recognition have already been produced[22,28–31]. Nonetheless, base editors are now well-tested for use in non-human primate embryos, which are extremely valuable for disease modeling and therapeutic development. In this study, we directly tested 4 base editors at 11 genes/loci in cynomolgus monkey embryos. We obtained the best results from SaKKH-BE3 for C-to-T conversions and ABE7.10 for A-to-G conversions. Although we only tested SaKKH-BE3 in two genes, this BE3 variant worked extremely well compared to SpCas9-BE3 or -BE4: all the embryos had C-to-T conversions with very high frequency (near 100% C-to-T conversion in 13 out of 16 edited cases), almost 100% C-to-T conversion accuracy (only 1 out of 48 edited Cs was converted incorrectly), and no indels. Although results may vary at other genes/loci and should be tested further, we suggest SaKKH-BE3 as the first choice for C-to-T base editing in monkey embryos. Of all editors that we tested, ABE showed the highest conversion accuracy (27/27 As converted to Gs with no indels), an important feature for gene therapy since incorrect editing can result in additional, potentially worse mutations. However, in contrast to its high accuracy, the conversion efficiency of ABE7.10 was relatively low and variable between genes and embryos, suggesting a need for further improvement of adenine base editors. Nonetheless, the work presented here demonstrates the advantages of ABEs, as editing at the *TP53 locus* by ABE outperformed a traditional HDR approach that we employed in a previous study in both efficiency and accuracy[15].

In BE3 and ABE, mammalian codon optimization of Cas9 and removal of potential poly A signal sites within Cas9 were recently reported to drastically increase conversion efficiency[28,29]. Because we used mRNA for injection, the same strategy may also improve A-to-G conversion efficiency by ABE in monkey embryos. Our success of highly efficient and precise C-to-T and A-to-G base editing by using BE3 variant and ABE enables introduction and potential restoration of a significant number of human pathogenic SNVs in cynomolgus monkey embryos and can accelerate monkey model production for human diseases.

To overcome the limitations imposed by the long reproductive cycle of cynomolgus monkeys, we successfully showed the feasibility of multiplexed base editing with BEs and/or ABEs in cynomolgus monkey embryos. We observed multiplex base editing in up to three sites not only in whole embryos and, most importantly, in single embryo cells (blastomeres). By blastomere genotyping, we also found significant mosaicism in all the multi-edited embryos, which could be due to the use of mRNA instead of ribonucleoprotein complexes (RNPs) and the late timing of injection. It is reported that the direct injection of CRISPR as an RNP[32–34], which acts fast and efficiently on the target site, can eliminate mosaicism when paired with injection into MII

oocytes[35]. Alternatively, injection of base editor mRNA and sgRNA into two- or four-cell stage human embryos much improved conversion efficiency, and thus reduced mosaicism, making one third of injected human embryos (5 out of 15) non-mosaic 100% base edited[36]. Taken together, production and direct delivery of recombinant base editor protein[7] would be a major step forward toward non-mosaic embryos.

Obviously, obtaining non-mosaic founder monkeys is the ultimate goal of genome editing in non-human primates. However, in many cases mosaic founder monkeys are still useful for initial phenotypic analysis before production of second-generation non-mosaic monkeys, as demonstrated by recent studies of founder mutant monkeys with *MECP2* (ref. [37]) and *SHANK3* (ref. [12]) mutations. We recently generated *SHANK3* knockout monkeys including two non-mosaic and three mosaic founders[12]. Importantly, we observed strong autism-related phenotypes in all founder monkeys, including heterozygous mosaics. These studies strongly suggest that mosaic founder monkeys generated by base editors could be useful for preliminary phenotypic analyses, given that in our current study, SaKKH-BE3 and ABE in a multiplexed condition yielded averages of 80–95% and 50–80% edited allele frequencies (within edited embryos), respectively. Nonetheless, it is critical to improve base editing with a goal of efficiently generating non-mosaic, cleanly edited founder monkeys as this will significantly reduce the time and cost of non-human primate studies of disease and facilitate the analysis of polygenic phenotypes.

Trio-based whole-genome sequencing analysis identified only 71 fetus-specific de novo variants, comprised of 66 SNVs and 5 indels. The number of de novo variants observed is in the range of naturally occurring germline de novo SNVs per generation in cynomolgus macaque embryos. The lack of overlap between these de novo variants and the predicted *FAH* sgRNA off-target sites suggests that base editors do not induce guide RNA-dependent off-target mutations, consistent with previous whole-genome sequencing studies in mouse[38] and human[36] embryos edited with CBEs. The use of SaKKH-Cas9, which recognizes a longer PAM and thus reduces guide RNA-dependent off-target mutations, will further increase the utility of base editing not only by its accurate base editing capacity, but also its lower capacity for RNA-dependent off-target mutation. However, previous studies have shown that guide RNA-independent deaminase-induced random genomic mutation is a significant concern with CBE-, but not ABE-mediated base editing. These reports showed a 20-fold increase in genome-wide mutation rates (average 283 SNVs in BE3-treated cells per generation compared to average 14 SNVs in control cells derived from the same mouse embryo) by GOTI (Genome-wide Off-target analysis by Two-cell embryo Injection)[26], and a 2-fold increase (average 221 SNVs in BE3-treated mice compared to average 132 SNVs in control mice) by large scale trio-based whole-genome sequencing[38], mainly due to increased C-to-T and G-to-A conversions mediated by cytidine deaminase. We only detected 66 fetus-specific de novo SNVs, which is in the range of the naturally occurring spontaneous mutation rate in primates (22-78 per generation)[39–41]. Furthermore, our estimate of off target C-to-T and G-to-A is only 3.6. Thus, in our trio-based whole-genome sequencing experimental design, we did not detect obvious guide RNA-independent random C-to-T (or G-to-A) conversion by BE3. Taken together, these results suggest that base editing is highly specific in cynomolgus monkey embryos. Since we only analyzed a base edited fetus together with its parents, further whole-genome sequencing studies with more controls, such as untreated control monkeys or endogenous controls made by 2-cell injection strategies are very important to evaluate both Cas9-dependent and deaminase-dependent off-target mutations by base editing. Additionally, although guide RNA-independent deaminase-induced

random RNA off-target mutation has been reported in both CBE- and ABE-mediated base editing[42–44], we delivered CBE and ABE as mRNA that acts transiently, reducing the risk of severe effects from unwanted RNA editing. Future studies using improved engineered deaminases with reduced random DNA[45–47] and RNA[42–44] editing activities will only serve to further reduce these risks and improve base editing efforts.

In summary, we not only show efficient and precise targeted base editing in cynomolgus monkey zygotes by BEs and/or ABE mRNA microinjection, but also demonstrate its utility for multiplexed simultaneous targeting of up to three target genes/sites as a means of overcoming long reproduction times required for developing primate models of polygenic diseases. Further, we show high on-target specificities of BE3/ABE by genome-wide analysis. Our results demonstrate the feasibility of BE3/ABE-based strategies for the development of polygenic primate disease models and identify areas for future improvement of base editing technologies.

## Methods

**Animals**. The animal experiments in this study were conducted at Guangdong Landao Biotechnology Co. Ltd (Landao). The Landao is an Association for Assessment and Accreditation of Laboratory Animal Care accredited facility. All animal protocols were approved in advance by the Institutional Animal Care and Use Committee of Guangdong Landao Biotechnology Co. Ltd and South China Agricultural University.

**RNA preparation**. The pCMV-BE3 (#73021), pCMV-ABE7.10 (#102919), saKKH-BE3 (#85170), and BE4-Gam (#100806) plasmids were obtained from Addgene. These plasmids were digested with NotI (NEB, Cat. No. R0189S) and BE3, BE4-Gam, and ABE7.10 mRNAs were synthesized using mMESSAGE mMACHINE T7 Ultra kit (Life Technologies, Cat. No. AM1345). T7-sgRNA cassettes were synthesized by PCR using pX459 plasmid (Addgene, #62988) and primers containing spacer and T7 sequences (Supplementary Table 7) with *Premix Taq* (Takara, Cat. No. R003Q). The sgRNAs were synthesized using the MEGAshortscript T7 Transcription Kit (Life Technologies, Cat. No. AM1354). The mRNAs and sgRNAs were purified using the MEGAclear Kit (Life Technologies, Cat, No. AM1908).

**Microinjection**. Cynomolgus monkey zygotes were prepared as follows[48]. Monkey semen was collected from a male monkey (aged 11 years and body weight 9.8 kg) by penile electro-ejaculation. Four female monkeys (aged 5–9 years and body weight 3.6–5.1 kg) at menses were intramuscularly administered 1.4 µg rhFSH (recombinant human follitropin alfa; GONAL-F, Merck Serono) twice per day for eight days, followed by 80 µg rhCG (recombinant human chorionic gonadotropin alfa; OVIDREL, Merck Serono) on day 9. Cumulus-oocyte complexes were collected by laparoscopic follicular aspiration 33–36 hours after rhCG administration and oocytes were stripped of cumulus cells by pipetting after brief exposure (<1 min) to hyaluronidase (0.5 mg/ml) at 37 °C. Metaphase II (MII, first polar body is present) oocytes were cultured in CMRL-1066 medium containing 0.1% Na-lactate and 10% fetal bovine serum (FBS; Hyclone Laboratories, SH30088.02) and fertilized by intracytoplasmic sperm injection (ICSI). At 10–12 hours after ICSI, a mixture of base editor mRNA (100 ng/µl) and sgRNA (50 ng/µl) was injected into the cytoplasm of using an Eppendorf's microinjection system and a microinjector (FemtoJet 4i, Eppendorf). Injections were performed thrice for *FAH*-E4 and twice for the others. The zygotes were then cultured under mineral oil in hamster embryo culture medium-9 containing 10% fetal bovine serum at 37 °C in 5% $CO_2$, 5% $O_2$, and 90% $N_2$. The zygotes were cultured more than 3 days to collect for genotyping or embryo transfer. For transfer, surrogate mothers with two consecutive normal menstrual cycles (26–32 days), a freshly ovulated follicle, and uterus with normal echo in uterus cavity immediately before embryo transfer were selected as surrogate mothers. Four to seven embryos at the eight-cell to blastocyst stage were transferred into one oviduct of each recipient via laparoscopy with a fixed polythene catheter. Pregnancy in surrogate mothers was diagnosed after imaging an apparent conceptus with beating heart at 25 days post-transfer.

**Genotyping**. Genomic DNA was amplified from embryos using the REPLI-g Single Cell Kit (QIAGEN, Cat. No. 150345). PCR amplification was performed using *Ex Taq* (Takara, Cat. No. RR01AM) or PrimeSTAR GXL DNA Polymerase (Takara, Cat. No. R050A). The list of genotyping primers is shown in Supplementary Table 8. The PCR amplicons were gel purified and sequenced. Sanger sequence results were analyzed using SnapGene statistics were performed using Graphpad Prism 7. To analyze mutation frequencies, PCR products were subcloned using the pMD18-T Vector Cloning Kit (Takara, Cat. No. 6011) or Mighty TA-cloning Reagent Set for PrimeSTAR (Takara, Cat. No. 6019), and transformed into *E. coli*. Plasmids were purified and analyzed by Sanger sequencing. For blastomere genotyping, blastomeres were isolated from 4 to 8 cell stage embryos and genomic

DNA was amplified and analyzed using the same methods employed for embryo genotyping. For fetus genotyping, genomic DNA was extracted from various tissues of the mutant fetus and blood of the father and mother using TIANamp Genomic DNA Kit (Tiangen, DP304), then PCR amplification of *FAH* target site was performed and subjected for deep sequencing at Genewiz.

**T7E1 cleavage assay**. Targeted DNA fragments were amplified (Primers are shown in Supplementary Data 2) with PrimerSTAR HS DNA polymerase (Takara, Cat. No. R010A) and purified via DNA Fragment Purification Kit (Takara, Cat. No. 9761). Purified PCR amplicons were heat denatured, re-annealed, and then digested with T7E1 (NEB, M0302L) for 30 min at 37 °C and separated on 2% agarose gel.

**Whole-genome sequencing**. Genomic DNA was extracted from the liver of the mutant fetus and blood of the father and mother using TIANamp Genomic DNA Kit (Tiangen, DP304). Two-hundred nanograms of genomic DNA was subjected to whole-genome sequencing at a sequencing depth of 30–40× using an Illumina HiSeq 2000 platform at Novogene Bioinformatics Institute, Beijing, China. Reads were aligned to the reference genome using bwa-mem (bwa version 0.7.15-r1140; Macaca fascicularis genome assembly GCF_000364345.1) and duplicates were marked using PicardTools (version 1.111(1909)). To detect de novo mutation sites, the bam files from the mother, father and fetus were provided as input to GATK Haplotype Caller (GATK version gatk-4.1.3.0). Only autosomal chromosomes were included in the analysis. The raw SNV calls were filtered using the following thresholds: QD < 2.0 or FS > 60.0 or MQ < 40 or MQRankSum < −12.5 or ReadPosRankSum < −8.0 or QUAL < 100. For indels, the following filters were used: QD < 2.0 or FS > 200 or ReadPosRankSum < −20 or QUAL < 100. Except for the quality filter, the filter values were recommended by GATK Best Practices for non-model genome assemblies. We used a more stringent quality filter (QUAL < 100) since not all steps in the GATK Best Practices could be followed due to incomplete resources (e.g.-known high-quality SNP and indel sites). Regions with a high density of variant sites could be due to mapping issues, hence we filtered out sites that have at least 5 variants within a 100 bp window. We obtained 19,131,754 SNVs and 3,131,601 indels. To identify de novo variants, we followed the GATK Genotype Refinement workflow. First, we ran GATK's CalculateGenotypePosteriors where a PED file of the trio was provided and the option "–skip-population-priors" was used. Second, GATK's VariantFiltration was run to mark individuals with GQ < 30, and finally, GATK's VariantAnnotator was run with the option "-A PossibleDenovo". The Genotype Refinement workflow yielded 1031 High Confidence de novo SNVs and 165 High Confidence de novo indels. We performed another set of filtering to obtain the final set of de novo variants. First, we ran samtools mpileup to count the raw number of reads at each candidate de novo site. Each site must have at least 20 reads for the mother, father, and fetus (DP < 20), if parents are homozygous ref then the alternate allele must not be found in the parents, if parents are homozygous alt then the reference allele must not be found in the parents, and the allele balance of the fetus must be between 0.3 and 0.7. This yielded our final number of 66 de novo SNVs and 5 de novo indels. These were visualized by IGV (version 2.6.1). We computationally analyzed the overlap between these de novo mutations and in silico predicted potential off-target sites predicted by Benchling (https://benchling.com), CRISPOR[49], and Cas-OFFinder[50], and then manually confirmed. To calculate the expected number of de novo SNVs, we downloaded the *Cynomolgus macaque* variant sites from dbSNP which are now released via the European Variation Archive (EVA, https://www.ebi.ac.uk/eva). For each of the possible 9 base substitutions, we counted the number of dbSNP/EVA sites, divided the count by the total number of dbSNP/EVA sites, and multiplied the fraction with 66 to get the expected number of de novo SNVs (Supplementary Data 3).

**Reporting summary**. Further information on research design is available in the Nature Research Reporting Summary linked to this article.

## Data availability

All deep sequencing and whole-genome sequencing data from this study has been deposited at NCBI Sequence Read Archive (SRA) under accession numbers PRJNA561611 and PRJNA505503, respectively. The source data 603 underlying Supplementary Figs. 4a and 8a–l are provided as a Source data file. Data supporting the findings of this work are available within the paper and its Supplementary Information files. The datasets generated and analyzed during the current study are available from the corresponding author upon request.

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

## Acknowledgements

This work was supported by grants from Guangdong Special Support Program (2019BT02Y276), the National Key R&D Program of China (2018YFA0801404), the Frontier and Innovation of Key Technology Project in Science and Technology Department of Guangdong Province (2014B020225007), Program for New Century Excellent Talents in University (NCET-12-1078), National Natural Science Foundation of China (81941006 and 81771579), South China Agricultural University Ph.D. students abroad (outside) joint training project (2018LHPY009), Rett Syndrome Research Trust, and the Hock E. Tan and K. Lisa Yang Center for Autism Research. We thank Baron Koylass (European Variation Archive, EMBL-EBI) for providing the Macaca fascicularis dbSNP data, and Qian Chen (MIT), Menglong Zeng (MIT) and Dheeraj Roy (Broad Institute) for helpful discussions.

## Author contributions

S.Y., G.F., T.A., and W.Z. designed and supervised the study. W.Z., C.D., Y.H., Y.T., D.X., Z.F., D.L., and Z.Z. performed experiments. R.D.R., T.A., W.Z., and M.J. performed whole-genome sequencing analysis. S.Y., G.F., T.A., W.Z., Y.Z., J.J.W., Z.B., Q.H. and X.Z. collected and analysed data. T.A., R.D.R., J.J.W., and W.Z. wrote the paper with the input from all the authors. All of the authors read and approved the final paper.

## Competing interests

The authors declare no competing interests.
