## [Peer Review File · Nature Communications]

Reviewers' Comments:

Reviewer #1:

Remarks to the Author:

Manuscript: Multiplex precise base editing in cynomolgus monkeys (NCOMMS-18-34407)

In this manuscript, Zhang and colleagues demonstrate the use of CRISPR-based cytidine- and/or adenine- base-editors to produce precise and quite efficient simultaneous single, double and triple multiplex DNA base editing of up to three target sites across 11 genes/loci in cynomolgus monkey embryos. These results are exciting in that they constitute the first steps necessary to develop techniques able to model polygenic diseases in nonhuman primates. They also give great hope to the development of methods for simultaneously correcting multiple single-nucleotide variants (SNVs) that are typical of polygenic diseases, greatly expanding the potential of gene therapy approaches.

As exciting as the report is, however, results are quite preliminary in nature, as they do not show live births of the modified embryos, and they do not show phenotypes associated with the mutations produced. There's a lot of effort placed on the optimization of base editors to produce the intended mutations as precisely and efficiently as possible, without however providing some rationale for why these mutations were chosen (e.g. are cynomolgus monkeys the best animal model for Alzheimer's disease, or for breast cancer?), and without an actual demonstration that the modified embryos are viable. Furthermore, there was no mention of how the implanted embryos were screened non-destructively prior to being implanted into surrogate mothers. These requirements are essential as far as the production of genetically-engineered nonhuman primates go.

I have the following concerns in need of further discussion:

1. FAH E4 and APP E2 mutations: These were the most efficient mutations, using either BE3 (FAH) or ABE (APP). It is mentioned that embryos were cultured for up to 3 days or transferred into surrogate mothers. What is the status of the embryos transferred into surrogate mothers? Why is this data not shown here? While it is fantastic that the efficiency of mutations was so high, it is equally important that the mutations allow viable embryos to fully develop and be born.
2. Embryo transfer into surrogate mothers: while it was mentioned that some embryos with the single mutation of FAH were transferred into surrogate mothers, the transfer of double- and triple-edited embryos was not mentioned. Presumably the efficiency of double and triple mutations decreases to the point that it is not advisable to transfer the embryos. Was there any attempt to transfer those embryos and what was the rationale for doing (or not) so? And the embryos that were implanted into surrogate mothers, were they screened for the intended mutations prior to embryo transfer?
3. Array of Mutations/ Target Genes: In this study a number of different target genes and mutations are used, including FAH, APP, HBB, TP53, EMX1, FANCF and BRCA1. Why were all these genes targeted? Are these all genetically-engineered animals that are in the pipeline to be produced? Some rationale for targeting these genes should be provided.

Reviewer #2:

Remarks to the Author:

The authors demonstrated high efficiency multiplexed base editing to install disease-relevant mutations in Cynomolgus Monkey (Macaque) embryos using BE3, BE4-Gam, saKKH BE3, and ABE7.10. To model complex, multigenic diseases in this important preclinical model, multiplexed and precise editing is important, since the reproductive cycle of the macaque is prohibitively long (~5 years) for crossing animals to obtain double mutants. The authors demonstrate relevantly high efficiency base editing in macaque embryos targeting up to three sites with simultaneous use

of CBE and ABE and investigate off-target editing. mRNA encoding editor(s) and guide(s) was microinjected into embryos at the zygote stage closely following fertilization. This study provides a proof of concept and useful reference for others attempting to make SNP disease models in macaque with base editors, and would be a valuable contribution to Nature Communications and an important milestone for the field. It could be improved by clarifying the presentation of the editing outcomes and performing a more thorough analysis of off-targets to be of most use to the field.

Major comments:

- Overall, the way that the results were described could have been more clear and representative of the data. The authors' statements are not false but the way results are presented is not as clear as possible (e.g., "substitution efficiency" defined as clones that have some form of C to T editing, regardless of indels or unwanted mutations). The fact that two of the embryos seem to be completely and cleanly edited with BE3 editor and others with saKKH BE3 nearly completely edited is impressive and represents a useful milestone for the field.
- Mosaicism was addressed in this paper by measuring editing efficiency in single embryos, but aside from alternative editors, no other methods were tested or suggested for how to reduce mosaicism. This is a major shortcoming of the study, since backcrossing would still need to be used to obtain non-mosaic animals.
- This paper lacked comparisons of their base editing data with other methods used in the field/other available methods. This is understandable, though, since macaque editing is so resource intensive.
- Choice of editor was based on performance in embryos at very few sites. Other groups should probably still screen editors at their site of interest, as was stated by the authors.
- Health/appearance/viability of embryos and implanted monkeys not reported
- Multiplexed and precise editing for modeling of complex diseases seems reasonable with the long reproductive cycle of macaque, but the examples given of multiplexed editing (beta thalassemia and cancer) failed to justify the need for the solution to this problem. While this paper's importance is in technology development, more meaningful sites to install in a multiplexed manner for disease modeling would make the study stronger.
- The off-target analysis is not sufficiently thorough to determine if base editing is precise in macaque embryos.
- The authors should provide justification for and discussion of the rates of error associated with the methods used.

Results Section 1: Single C-to-T base editing in monkey embryos

Targeting FAH with BE3 in macaque embryos by microinjection of mRNA and guide.

BE4-Gam tested at the same FAH guide increases product cleanliness but also reduces on-target efficiency substantially.

Fig 1b. implies two injections? From the methods, it seems that one injection of editor and guide was done.

Fig 1c,e. Hard to visualize with slightly different shades of green—put in a bar graph format or add percent values?

Fig 1i. The APP locus was not cleanly edited at 100% efficiency. Of edited embryos, none were completely edited (some clones in each embryo remain unedited). This table is not particularly informative and should be made more clear.

Supplementary Figure 1

Many clones carry what seems to be a C to A mutation (or some other unexpected editing). The authors only mention this in figure 1d. Clear presentation of the percentage of cleanly edited clones in each embryo would be informative.

Section 2: Single A to G base editing in monkey embryos

Target amyloid precursor protein with ABE 7.10 in macaque embryos by microinjection of mRNA

and guide.

Again, rephrasing of some of the editing percentage metrics to be more precise and clear would be better. Statements made referring to analysis of data are not false, but should be stated more clearly. Repeatedly, "X percent of embryos had the A to G conversion at the target sites" is not forthcoming about the fact that none were completely edited, and that the distribution of editing outcomes contain clones w/ multiple A's in the window edited. The authors do, in this section and the last, include indel percentages and non C to T or A to G mutation percentages.

Section 3: Double/triple multiplex C-to-T or A-to-G base editing in monkey embryos

Editing w/ CBE or ABE in macaque embryos by microinjection of mRNA and guide. For CBE targets two or three guides were targeted to FAH gene (Chr 7, across different exons, first pair 4000bp apart). All guides target the same strand of DNA.

Line 151: "...almost 100%" I can't determine from the data presented in the main text or SI why embryo #9 is not 100% C-to-T. Is this due to inadequate coverage of alleles?

Again, redefining of the "A-to-G editing efficiency" to not be "The # of embryos that have a clone with some sort of A-to-G edit at the locus" would be more informative.

The multiplexed mutations installed in this section cause cancer and beta thalassemia. These embryos were not implanted but while the sites are relevant on their own, their combination is not particularly compelling.

To measure editing in single cells, long PCR amplicons were run through both edited sites on FAH (4kb). This seems to be a reasonable way at getting at editing in same cells.

The rate of mutation incorporation during whole genome amplification (WGA) for editing determination should be addressed and justified.

Section 4: Triple multiplex C-to-T and A-to-G base editing in monkey embryos

Line 174: "G to A" should be A to G in this context (referring to both editors)

To multiplex CBE and ABE, the authors used saKKH CBE and spABE, which do not accept the other's guide and also have different PAM recognition sequences. Editing was observed in a high proportion of embryos with all three sites.

Single blastomere sequencing (Table in Fig 3e)—where is the allele table/breakdown as there has been for the other editing experiments?

Section 5: Off-target analyses

Whole genome sequencing at 30-40x sequencing depth was carried out on fetal liver tissue to identify off-target affects in a trio-based manner (comparing paternal and maternal SNPs from blood to those of the fetal liver).

Fig 4a. W->Y mutation results from a C to T mutation and adjacent C to A. This is an important anomaly that should be addressed.

The off-target analysis needs to be more thorough to determine if base editing is precise in macaque embryos. No off targets were reported. While this is not necessarily false, off-target analysis needs more complete than the authors offered in this study:

- o Off target analysis should be done with multiple organs—since mosaicism seems plausible.
- o Suggestion: A good control for the off-target study could be injection of base editor into one cell at the two-cell stage, then WGS after some time.
- o How many cells were input for WGS?
- o What is the rate of random mutations? More explanation is needed for how the fetus-only SNPs were binned (ex: of the mutations that were not present in the parents, how was a "PAM containing site" determined?)
- o Off-targets cannot be defined simply as sites w/ a predicted PAM and "expected" C to T mutations.

Detailed response to reviewers' comments:

We sincerely thank the reviewers for their encouraging and constructive comments to strengthen our study. As detailed below, we have followed the reviewers' suggestions carefully and revised the manuscript accordingly.

Reviewer #1 (Remarks to the Author):

Manuscript: Multiplex precise base editing in cynomolgus monkeys (NCOMMS-18-34407)

In this manuscript, Zhang and colleagues demonstrate the use of CRISPR-based cytidine- and/or adenine- base-editors to produce precise and quite efficient simultaneous single, double and triple multiplex DNA base editing of up to three target sites across 11 genes/loci in cynomolgus monkey embryos. These results are exciting in that they constitute the first steps necessary to develop techniques able to model polygenic diseases in nonhuman primates. They also give great hope to the development of methods for simultaneously correcting multiple single-nucleotide variants (SNVs) that are typical of polygenic diseases, greatly expanding the potential of gene therapy approaches.

As exciting as the report is, however, results are quite preliminary in nature, as they do not show live births of the modified embryos, and they do not show phenotypes associated with the mutations produced. There's a lot of effort placed on the optimization of base editors to produce the intended mutations as precisely and efficiently as possible, without however providing some rationale for why these mutations were chosen (e.g. are cynomolgus monkeys the best animal model for Alzheimer's disease, or for breast cancer?), and without an actual demonstration that the modified embryos are viable. Furthermore, there was no mention of how the implanted embryos were screened non-destructively prior to being implanted into surrogate mothers. These requirements are essential as far as the production of genetically-engineered nonhuman primates go.

We are thankful for the encouraging and insightful comments. As detailed below, we made every effort to address the noted concerns and performed new analyses to improve the manuscript. We agree with the comment on the importance of production of live mutant monkeys and phenotypic analysis of the induced mutations. However, we designed this study as a proof-of-concept to investigate the utility of CRISPR base editing and show the feasibility of multiplex introduction of precise mutations in embryos of cynomolgus macaque monkey. It can take years to produce live monkeys due to the long gestation period of cynomolgus macaque and the highly limited number of egg donors and surrogate mothers. Also, as seen in our study, successful pregnancies are often lost due to miscarriage, lengthening the amount of time necessary to produce live animals. Furthermore, the phenotypic analysis of the induced mutations adds even more time. We discussed these points with editor about these points and agreed not to include these data in the current manuscript. We would like to show live mutant monkey production and phenotypic analysis in our next study.

Regarding target choice for multiplex editing, these genes are all related to various diseases (cancer, tyrosinemia, Fanconi Anemia, Alzheimer's disease, and Kallmann Syndrome) that are in our list for monkey model production. While the mutations installed are not associated with the same disease, using them as proofs-of-principle for this study allowed us to both test the feasibility of multiplex base editing and test the ability of base editing to install mutations that we are already planning on generating in our lab.

For embryo transfer, we did not perform prescreening of embryos to identify those that were correctly modified.

I have the following concerns in need of further discussion:

1. FAH E4 and APP E2 mutations: These were the most efficient mutations, using either BE3 (FAH) or ABE (APP). It is mentioned that embryos were cultured for up to 3 days or transferred into surrogate mothers. What is the status of the embryos transferred into surrogate mothers? Why is this data not shown here? While it is fantastic that the efficiency of mutations was so high, it is equally important that the mutations allow viable embryos to fully develop and be born.

We apologize that we did not include the detail of embryo transfer to surrogate mothers in our initial manuscript. Now we included the detail, and further added the comparison of embryo development (**new Supplementary Figs. 12-13**). The rates of embryo development until blastocyst stage were comparable between buffer-injected controls and the experimental group injected with BE3 mRNA and sgRNA targeting *FAH*. We performed transfer of embryos injected with BE3 and *FAH* exon 4 sgRNA once, and this data is now included in the last section of the revised manuscript. We used embryos developed at

normal timing with symmetric blastomeres for transfer. Unfortunately, we could not obtain live births due to miscarriages after collection of amniotic fluids from pregnant recipients.

We agree with the comment on the importance of production of live mutant monkeys. However, again, we designed this study as a proof-of-concept to investigate the feasibility of multiplex base editing in monkey embryos. In addition, it takes years to produce live monkeys due to the long gestation period of cynomolgus macaque and the highly limited number of egg donors and surrogate mothers. We would like to show this point in our next study, as we feel that these downstream analyses are beyond the scope of this study.

2. Embryo transfer into surrogate mothers: while it was mentioned that some embryos with the single mutation of FAH were transferred into surrogate mothers, the transfer of double- and triple-edited embryos was not mentioned. Presumably the efficiency of double and triple mutations decreases to the point that it is not advisable to transfer the embryos. Was there any attempt to transfer those embryos and what was the rationale for doing (or not) so? And the embryos that were implanted into surrogate mothers, were they screened for the intended mutations prior to embryo transfer?

No, we did not transfer the embryos injected with multiple guideRNAs/base editors. Again, we designed this study as a proof-of-concept to investigate the feasibility of multiplex base editing in monkey embryos, we only tried the embryo transfer with single *FAH*-sgRNA/BE3 injection once due to the limited cynomolgus monkey resources. These embryos were not screened/selected before transfer.

3. Array of Mutations/ Target Genes: In this study a number of different target genes and mutations are used, including FAH, APP, HBB, TP53, EMX1, FANCF and BRCA1.

Why were all these genes targeted? Are these all genetically-engineered animals that are in the pipeline to be produced? Some rationale for targeting these genes should be provided.

Yes, these genes are all related to various diseases (cancer, tyrosinemia, Fanconi Anemia, Alzheimer's disease, and Kallmann Syndrome), and are in our list for monkey model production.

Reviewer #2 (Remarks to the Author):

The authors demonstrated high efficiency multiplexed base editing to install disease-relevant mutations in Cynomolgus Monkey (Macaque) embryos using BE3, BE4-Gam, saKKH BE3, and ABE7.10. To model complex, multigenic diseases in this important preclinical model, multiplexed and precise editing is important, since the reproductive cycle of the macaque is prohibitively long (~5 years) for crossing animals to obtain double mutants. The authors demonstrate relevantly high efficiency base editing in macaque embryos targeting up to three sites with simultaneous use of CBE and ABE and investigate off-target editing. mRNA encoding editor(s) and guide(s) was microinjected into embryos at the zygote stage closely following fertilization. This study provides a proof of concept and useful reference for others attempting to make SNP disease models in macaque with base editors, and would be a valuable contribution to Nature Communications and an important milestone for the field. It could be improved by clarifying the presentation of the editing outcomes and performing a more thorough analysis of off-targets to be of most use to the field.

We are thankful for the constructive comments to improve the manuscript. As detailed below, we made every effort to address the concerns and performed new analyses to improve the manuscript.

Major comments:

· Overall, the way that the results were described could have been more clear and representative of the data. The authors' statements are not false but the way results are presented is not as clear as possible (e.g.,: "substitution efficiency" defined as clones that have some form of C to T editing, regardless of indels or unwanted mutations). The fact that two of the embryos seem to be completely and cleanly edited with BE3 editor and others with saKKH BE3 nearly completely edited is impressive and represents a useful milestone for the field.

The presentation of the editing outcomes is now improved throughout the manuscript. To further clarify the editing outcomes in each embryos, we added the following information: 1) In addition to the color gradient representing conversion frequency in the figures, we included actual percent values of the frequencies of intended conversions (C-to-T or A-to-G) and unintended conversions (non-C-to-T or non-A-to-G) at each single base. We also included actual percent values of indels as well (**new Figs. 1c-d, 2b-d, 3b**). 2) The unintended conversions are now shown as dot graphs to indicate the editing outcomes in each embryo, rather than as an average (**new Figs. 1e-f, 2e-g, 3c-d**). Also, indels are moved to main figures from supplementary figures and now shown as dot graphs (**new Figs. 1g, 2e-g, 3e**). 3) In the main text, we rephrased and described the details of editing outcomes including unintended conversions (non-C-to-T or non-A-to-G) and indels.

· Mosaicism was addressed in this paper by measuring editing efficiency in single embryos, but aside from alternative editors, no other methods were tested or suggested for how to reduce mosaicism. This is a major shortcoming of the study, since backcrossing would still need to be used to obtain non-mosaic animals.

This is an important point and we're grateful that it was brought to our attention. Although we did single blastomere analysis to address whether multiplex base editing truly occurred in a single blastomere, we also found mosaic editing outcomes in each embryo. As the reviewer pointed out, mosaicism is a serious issue in for the production of genetically modified monkey model according to long reproduction cycle of cynomolgus monkey for backcrossing. We therefore added discussion about potential ways to reduce mosaicism.

· This paper lacked comparisons of their base editing data with other methods used in the field/other available methods. This is understandable, though, since macaque editing is so resource intensive.

We thank the reviewer for understanding the limitation of resource in macaque research. Although we couldn't directly compare base editing and other standard methods such as DNA double strand break (DSB)-stimulated homology-mediated repair (HDR) approach in this study, we previously reported CRISPR-mediated HDR approach at the *TP53* gene in cynomolgus monkey embryos using wildtype Cas9, sgRNA, and single-strand oligodeoxynucleotides (ssODN) (Wan H *et al.*, One-step generation of p53 gene biallelic mutant Cynomolgus monkey via the CRISPR/Cas system. *Cell Res* 25:258-261 (2015)). In this previous report, we found that 2 out of 9 embryos had HDR allele; Embryo 1 had 6 PCR clones of HDR out of total 12 PCR clones sequenced, although all 6 HDR clones had an additional unintended 1bp deletion. The other 6 non-HDR clones contained a 1bp

insertion. Thus, embryo 1 did not have any precise HDR alleles. Embryo 2 had a precise HDR allele, but its frequency was just 7.7% - 2 PCR clones out of total 26 PCR clones sequenced. The other 24 clones carried 9bp and 232bp deletions. Thus, our conclusion at that time was that the standard knockin approach with CRISPR and ssODN in cynomolgus monkey embryos is inefficient and imprecise, and thus, not suitable for knockin monkey production carrying specific human mutations, nor modeling polygenic diseases by installing multiple substitutions at the same time across genes. In the current manuscript, *TP53* was also targeted by ABE together with *HBB* and all 5 embryos had intended A-to-G conversions with frequencies of individual embryos ranging from 8.3% to 52% (average 32.1%). Importantly, we didn't observe non-A-to-G substitutions or indels (**new Figs 2d, g, h**). Thus, we believe that base editing outperforms the standard HDR approaches for precise and efficient nucleotide conversions in cynomolgus monkey embryos. We added these points to the discussion in the revised manuscript.

· Choice of editor was based on performance in embryos at very few sites. Other groups should probably still screen editors at their site of interest, as was stated by the authors.

As is pointed out and discussed, we tested the performances of several base editors at a limited number of target sites/genes, thus we and others will still need to test the best editor and sgRNA for the sites of interest. However, we believe that our success with SaKKH-BE3 at 2 genes and ABE at 4 genes provides a good benchmark for base editing in monkey embryos and a starting point for others looking to use base editing for monkey model production, which will accelerate preclinical study.

· Health/appearance/viability of embryos and implanted monkeys not reported

Please see our response to the reviewer #1's comment 1 above (also pasted here).

We apologize that we did not include the detail of embryo transfer to surrogate mothers in our initial manuscript. Now we included the detail, and further added the comparison of embryo development (**new Supplementary Figs. 12-13**). The rates of embryo development until blastocyst stage were comparable between buffer-injected controls and the experimental group injected with BE3 mRNA and sgRNA targeting *FAH*. We performed transfer of embryos injected with BE3 and *FAH* exon 4 sgRNA once, and this data is now included in the last section of the revised manuscript. We used embryos developed at normal timing with symmetric blastomeres for transfer. Unfortunately, we could not obtain live births due to miscarriages after collection of amniotic fluids from pregnant recipients.

· Multiplexed and precise editing for modeling of complex diseases seems reasonable with the long reproductive cycle of macaque, but the examples given of multiplexed editing (beta thalassemia and cancer) failed to justify the need for the solution to this problem. While this paper's importance is in technology development, more meaningful sites to install in a multiplexed manner for disease modeling would make the study stronger.

We thank the reviewer for pointing out this important point and apologize that we didn't provide enough scientific justifications regarding target choices for multiplex base editing. These target genes are all related to various diseases and in our list for production of genetically modified monkey models carrying human disease mutations using base editors. We designed this study as a proof-of-concept to investigate the utility of CRISPR base editing and show the feasibility of multiplex introduction of precise mutations in embryos of cynomolgus macaque monkey. While the mutations installed are not associated with the

same disease, using them as proofs-of-principle for this study allowed us to both test the feasibility of multiplex base editing and test the ability of base editing to install mutations that we are already planning on generating in our lab. Based on these results, future studies to generate and analyze specific polygenic disorders can be designed and carried out.

· The off-target analysis is not sufficiently thorough to determine if base editing is precise in macaque embryos.

We thank the reviewer for pointing out this important point. Since off-target editing has become more important problem in base editing during this revision, we entirely reanalyzed our whole genome sequencing (WGS) data to list both sgRNA/Cas9-related and deaminase-related off-target mutation in an unbiased manner in collaboration with our bioinformatics experts. We applied the well-established GATK pipeline on the WGS data to detect de novo SNVs. The detail of analysis is added into the Methods section. As a result, we found 71 high-quality *de novo* variants (including 66 SNVs and 5 indels) in the *FAH* mutant fetus that were not found in its parental genomes. We did not observe any overlap between these de novo variations and potential *FAH* sgRNA off-target sites predicted by three algorithms, indicating a low probability of Cas9-mediated guideRNA-dependent off-target mutations. Recently, two papers reported CBE-induced, guideRNA sequence-independent, deaminase-mediated C-to-T mutations in both mouse and rice. Although we could not distinguish deaminase-mediated mutations from *de novo* variations based on our trio-based WGS design, the relatively small number of fetus-specific variations was within the range of naturally occurring *de novo* variation and suggests deaminase-mediated random C-to-T mutations are not present at detectable levels in this *FAH* mutant cynomolgus macaque fetus.

· The authors should provide justification for and discussion of the rates of error associated with the methods used.

We thank the reviewer for pointing out this important point to ensure the accuracy of our genotyping. To address this question, we collected uninjected wildtype embryos and amplified genomic DNA by whole genome amplification (WGA). Then, all of the target sites we used in this manuscript were PCR amplified, cloned, and sequenced as same as we did for injected embryos. We did not see any genetic modification in the target sites at all, suggesting that the error rate of the methods we used (WGA, PCR, cloning, Sanger sequences, data analysis) is low, and thus, our genotyping results are reliable (**new Supplementary Fig. 1**).

Results Section 1: Single C-to-T base editing in monkey embryos

Targeting FAH with BE3 in macaque embryos by microinjection of mRNA and guide.

BE4-Gam tested at the same FAH guide increases product cleanliness but also reduces on-target efficiency substantially.

Fig 1b. implies two injections? From the methods, it seems that one injection of editor and guide was done.

We thank the reviewer for pointing out this error and apologize for this. In the left injection scheme in the previous Fig. 1, we intended to show intracytoplasmic sperm injection

(ICSI). Thus, sperm was injected into MII oocytes, not “BE3 or ABE”. We corrected this in the revised figure (**New Fig. 1b**)

Fig 1c,e. Hard to visualize with slightly different shades of green—put in a bar graph format or add percent values?

We apologize for this unclear presentation of the data. We added percent values into the shaded area, which is now split into each target nucleotide as well as desired/undesired outcomes (e.g. C-to-T, non-C-to-T, and Indel).

Fig 1i. The APP locus was not cleanly edited at 100% efficiency. Of edited embryos, none were completely edited (some clones in each embryo remain unedited). This table is not particularly informative and should be made more clear.

We thank the reviewer for pointing out this point. To further clarify the editing outcome, we included the rate of clean editing at 100% efficiency with intended conversions (**New Tables 1-3**, based on Nature Communications instruction, we removed tables from the figures).

Supplementary Figure 1

Many clones carry what seems to be a C to A mutation (or some other unexpected editing). The authors only mention this in figure 1d. Clear presentation of the percentage of cleanly edited clones in each embryo would be informative.

We thank the reviewer for pointing out this important point. Yes, by BE3 treatment, we found many non-C-to-T edits, including C-to-A changes. To clarify these unintended changes, we added all the details of each embryo including clean editing, unintended editing, and indels into the figures.

Section 2: Single A to G base editing in monkey embryos

Target amyloid precursor protein with ABE 7.10 in macaque embryos by microinjection of mRNA and guide.

Again, rephrasing of some of the editing percentage metrics to be more precise and clear would be better. Statements made referring to analysis of data are not false, but should be stated more clearly. Repeatedly, “X percent of embryos had the A to G conversion at the target sites” is not forthcoming about the fact that none were completely edited, and that the distribution of editing outcomes contain clones w/ multiple A’s in the window edited. The authors do, in this section and the last, include indel percentages and non C to T or A to G mutation percentages.

Again, we thank the reviewer for pointing out the presentation and description of the editing outcomes, which is now improved throughout the manuscript. As we discussed above, to further clarify the editing outcomes in each embryos, we revised text to include allele frequencies, bystander conversions, unintended conversions (non-A-to-G), and indels.

Section 3: Double/triple multiplex C-to-T or A-to-G base editing in monkey embryos

Editing w/ CBE or ABE in macaque embryos by microinjection of mRNA and guide. For CBE targets two or three guides were targeted to FAH gene (Chr 7, across different exons, first pair 4000bp apart). All guides target the same strand of DNA.

Line 151: "...almost 100%" I can't determine from the data presented in the main text or SI why embryo #9 is not 100% C-to-T. Is this due to inadequate coverage of alleles?

As the reviewer pointed out, embryo #9 was actually 100% C-to-T edited at three loci. Since the numbers of sequenced clones were not so large (33, 35, and 40), we rephrased it as "100% within sequenced clones".

Again, redefining of the "A-to-G editing efficiency" to not be "The # of embryos that have a clone with some sort of A-to-G edit at the locus" would be more informative.

Again, we thank the reviewer for pointing out the description of the editing outcomes. As we discussed above, it is now improved to be more informative throughout the manuscript.

The multiplexed mutations installed in this section cause cancer and beta thalassemia. These embryos were not implanted but while the sites are relevant on their own, their combination is not particularly compelling.

Please see our response to the reviewers #1 and #2's comments above. (also pasted here)

We thank the reviewer for pointing out this important point and apologize that we didn't provide enough scientific justifications regarding target choices for multiplex base editing. These target genes are all related to various diseases and in our list for production of genetically modified monkey models carrying human disease mutations using base editors.

Again, we designed this study as a proof-of-concept to investigate the utility of CRISPR base editing and show the feasibility of multiplex introduction of precise mutations in embryos of cynomolgus macaque monkey. We are sure that, in our next study, we will design scientifically and preclinically meaningful human mutations to model polygenic diseases in cynomolgus monkey.

Regarding target choice for multiplex editing, these genes are all related to various diseases (cancer, tyrosinemia, Fanconi Anemia, Alzheimer's disease, and Kallmann Syndrome) that are in our list for monkey model production. While the mutations installed are not associated with the same disease, using them as proofs-of-principle for this study allowed us to both test the feasibility of multiplex base editing and test the ability of base editing to install mutations that we are already planning on generating in our lab.

To measure editing in single cells, long PCR amplicons were run through both edited sites on FAH (4kb). This seems to be a reasonable way at getting at editing in same cells.

We thank the reviewer for evaluation of this point. We believe this is a good way to show multiplex editing within a single cell.

The rate of mutation incorporation during whole genome amplification (WGA) for editing determination should be addressed and justified.

Please see our response to the reviewer #2's major comment above. (also pasted here)

We thank the reviewer for pointing out this important point to ensure the accuracy of our genotyping. To address this question, we collected uninjected wildtype embryos and amplified genomic DNA by whole genome amplification (WGA). Then, all of the target

sites we used in this manuscript were PCR amplified, cloned, and sequenced as same as we did for injected embryos. We did not see any genetic modification in the target sites at all, suggesting that the error rate of the methods we used (WGA, PCR, cloning, Sanger sequences, data analysis) is low, and thus, our genotyping results are reliable (**new Supplementary Fig. 1**).

Section 4: Triple multiplex C-to-T and A-to-G base editing in monkey embryos

Line 174: “G to A” should be A to G in this context (referring to both editors)

We thank the reviewer for pointing out this typo and apologize for this. As pointed out, this should be “A-to-G.” We corrected this in the revised manuscript.

To multiplex CBE and ABE, the authors used saKKH CBE and spABE, which do not accept the other’s guide and also have different PAM recognition sequences. Editing was observed in a high proportion of embryos with all three sites.

Yes, that’s correct. Fortunately, we found extremely efficient and clean editing performance with SaKKH-CBE.

Single blastomere sequencing (Table in Fig 3e)—where is the allele table/breakdown as there has been for the other editing experiments?

We apologize that we did not include the detail of single blastomere genotyping. These results are shown in **new Figs. 3f, g** as well as in **new Figs 2g, h** for double/triple CBE or ABE editing. In this set of injections, all embryos had *EMXI* and *FANCF* editing by

SaKKH-CBE and half had additional *BRCA1* editing by SpABE, in consistent with **Figs. 3b-e**. At the blastomere level, 70% of blastomeres had *EMX1* and *FANCF* editing by SaKKH-CBE and 23% of blastomeres had additional *BRCA1* editing by SpABE (In total 93% of blastomeres had *EMX1* and *FANCF* editing by SaKKH-CBE). These results support multiplex base editing within a single cell. We also found mosaicism in the all the multiplex edited embryos (but, importantly, we had non-mosaic embryos at least for SaKKH-CBE multiplex base editing). We added detailed description and discussion of how we can potentially overcome this issue.

Section 5: Off-target analyses

Whole genome sequencing at 30-40x sequencing depth was carried out on fetal liver tissue to identify off-target affects in a trio-based manner (comparing paternal and maternal SNPs from blood to those of the fetal liver).

Fig 4a. W->Y mutation results from a C to T mutation and adjacent C to A. This is an important anomaly that should be addressed.

We thank the reviewer for pointing out this. We added description about this non-C-to-T editing. Since this transfer was performed before we tested SaKKH-CBE3 or ABE, imprecise editing could be induced, as seen by embryo genotyping (**Fig. 1**).

The off-target analysis needs to be more thorough to determine if base editing is precise in macaque embryos. No off targets were reported. While this is not necessarily false, off-target analysis needs more complete than the authors offered in this study:

Please see our response to the reviewer's major comment above. (also pasted here)

We thank the reviewer for pointing out this important point. Since off-target editing has become more important problem in base editing during this revision, we entirely reanalyzed our whole genome sequencing (WGS) data to list both sgRNA/Cas9-related and deaminase-related off-target mutation in an unbiased manner in collaboration with our bioinformatics experts. We applied the well-established GATK pipeline on the WGS data to detect de novo SNVs.. The detail of analysis is added into the Methods section. As a result, we found 71 high-quality *de novo* variants (including 66 SNVs and 5 indels) in the *FAH* mutant fetus that were not found in its parental genomes. We did not observe any overlap between these de novo variations and potential *FAH* sgRNA off-target sites predicted by three algorithms, indicating a low probability of Cas9-mediated guideRNA-dependent off-target mutations. Recently, two papers reported CBE-induced, guideRNA sequence-independent, deaminase-mediated C-to-T mutations in both mouse and rice. Although we could not distinguish deaminase-mediated mutations from *de novo* variations based on our trio-based WGS design, the relatively small number of fetus-specific variations was within the range of naturally occurring *de novo* variation and suggests deaminase-mediated random C-to-T mutations are not present at detectable levels in this *FAH* mutant cynomolgus macaque fetus.

- Off target analysis should be done with multiple organs—since mosaicism seems plausible.

We thank the reviewer for pointing out this important point. We believe this mutant fetus is a heterozygous mutant without mosaicism. Although Sanger sequencing-based clone analysis may have some variation of mutant allele frequency around 50% across 8 organs of the mutant fetus (**previous Fig. 4b**), we believe this was due to the sequencing depth (~50-100 reads) of clone-based Sanger sequencing. We additionally performed deep

sequencing of these PCR amplicons by Illumina sequencing with tens of thousands of reads in sequencing depth, and found approximately 50% frequency of mutant FAH allele in all the 8 tissues in this fetus.

- Suggestion: A good control for the off-target study could be injection of base editor into one cell at the two-cell stage, then WGS after some time.

We thank the reviewer for the constructive suggestion to improve the off-target analysis. The suggested method called GOTI (Genome-wide Off-target analysis by Two-cell embryo Injection) developed by Prof. Hui Yang's team (Zuo et al., Science 364:289, 2019) is an elegant and sensitive method to detect off-target mutation by making control cells within a single embryo, enabling exclusion of germline de novo mutations and off-target comparison between base editor-injected and control cells. We agree the usefulness of GOTI for identification of off-target mutations in mice. However, to make this endogenous control cells in cynomolgus macaque embryos, we need to develop and optimize many new systems including genetic labeling of one cell (similar to Ai9 reporter mice carrying ROSA26-CAG-loxP-stop-loxP-tdTomato used in Zuo et al paper). However, this (or something similar) is not available in cynomolgus macaque monkeys. We would also need to optimize two-cell stage injection, Cre injection, separation of labeled and unlabeled cells from mosaic monkey fetus by FACS sorting, and WGS condition using sorted cells in which the amount is small. Therefore, the difficulty of establishing such a systems in cynomolgus macaque monkey embryos makes such an experiment unfeasible at this time.

- How many cells were input for WGS?

We used 200ng of purified genomic DNA from a liver of mutant fetus, which corresponds to approximately 33,000 cells. We added this point to the methods section.

- What is the rate of random mutations? More explanation is needed for how the fetus-only SNPs were binned (ex: of the mutations that were not present in the parents, how was a “PAM containing site” determined?)

We thank the reviewer for pointing out this important point and apologize for our previous incomplete WGS data analysis. In the previous manuscript, potential off-target sites of *FAH* guideRNA were predicted by BLAST, giving us 178 candidate sites flanked with PAM sequences (we apologize that the previous figure legend for Fig. 4d “d Summary of on-/off-target sites of sgRNA targeting *FAH* exon 4. Total 178 sites, including 1 on-target site and 1, 2, 4, 13, 13 and 34 off-target sites with 2, 3, 4, 5, 6, 7 and 8 mismatches, respectively, were identified by whole-genome sequencing analysis” was not accurate). Then, these candidates were checked to see whether they overlapped with WGS data from father, mother, and the *FAH* mutant fetus, especially within 4-8 nucleotide target window of BE3 within 20nt spacer sequence of the off-target candidate sites. As a result, we did not detect any overlap, and thus concluded no off-target mutation by BE3.

Since this analysis was just focused on predicted off-target candidate sites within WGS data, and guideRNA independent off-target has become important issue in base editing during this revision, we entirely reanalyzed our WGS data to list both sgRNA/Cas9-related and deaminase-related off-target mutation in an unbiased manner in collaboration with our bioinformatics experts. As a result, we found 71 high-quality *de novo* variants including 66 SNVs and 5 indels in the edited *FAH* mutant fetus. None of these overlapped with predicted potential off-target sites called with three separate algorithms. The number of *de novo*

variants was small and in the range of the naturally occurring spontaneous mutation rate across primates (22-78 per generation). These results suggest that the random mutation rate is not elevated in this *FAH* mutant cynomolgus macaque fetus.

- Off-targets cannot be defined simply as sites w/ a predicted PAM and “expected” C to T mutations.

We thank the reviewer for pointing out this important point. In the revised manuscript, we re-analyzed all the SNVs in our WGS data in an unbiased and genome-wide manner.

Reviewers' Comments:

Reviewer #1:

Remarks to the Author:

Review of: "Multiplex precise base editing in cynomolgus monkeys" (NCOMMS- 18-34407A)

Summary: This is a manuscript on the use of simultaneous multiplex base editors to modify up to three target sites across 11 genes/loci in cynomolgus monkey embryos to model complex, multigenic diseases. Results show the high specificity of base editing and demonstrate the feasibility of multiplex base editing for modeling polygenic diseases in primate zygotes. The original manuscript was reviewed by 2 reviewers, who provided extensive comments and suggestions for improvements. The main concerns from Reviewer #1 were related to the preliminary nature of the results, in that no live births of the modified embryos nor phenotypes associated with the mutations were demonstrated. Other concerns were for the lack of rationale and justification for the chosen mutations, and lack of demonstration on how the implanted embryos were screened non-destructively prior to being implanted into surrogate mothers as an essential requirement for the production of genetically-engineered nonhuman primates. Reviewer #2 was mostly concerned about the lack of clarity in the presentation of the results and the lack of an analysis of off-target mutations.

The authors made a substantial revision to the original submission in response to the comments provided by both reviewers, performing new analyses and addressing all concerns raised. In result, the revised manuscript is much improved. In response to the main concerns by Reviewer #1, the authors argued – successfully in my opinion, that the present study was designed as a proof-of-concept to investigate the feasibility of multiplex introduction of precise CRISPR base editing mutations in embryos of cynomolgus monkey. As it will take several years to produce live monkeys, the demonstration of live births and the associated phenotypic analysis of the induced mutations is by necessity topics for a future study. Not only the proof-of-concept shown here is essential to moving forward with the production of mutant animals, the results already obtained must be shared with the scientific community, specifically to guide other researchers in achieving similar results, and especially to inform the global community of how carefully the authors have been to optimize the methodology as much as possible to ensure that any animal eventually produced will have the intended mutations and to optimize and minimize the number of animals used to produce the invaluable mutant animals. The authors addressed reviewer #1's concern about embryo transfers well, including detailed information about the embryo transfers and their development relative to embryos that were injected with buffers. In response to the concern by Reviewer #1 that no screening of pre-implantation embryos was presented, the authors answered that only embryos with single mutations were implanted, but no prescreening of such embryos to identify mutations was performed. This is because there are no non-destructive methods to screen NHP embryos available. All genotypic confirmation needs to be performed after birth. As reviewer #1 correctly points out, "the efficiency of double and triple mutations decreases to the point that it is not advisable to transfer the embryos", and indeed the authors did not transfer any embryos with multiple mutations.

Reviewer #2 was strongly supportive of publication of the manuscript in Nature Communications, stating the manuscript will be an important milestone in the field. Reviewer #2 raised some concerns about the clarity of data presentation, and in particular, the lack of an effort by the authors to identify methods to reduce mosaicism. In response, the authors reorganized data presentation and added discussion about potential ways to reduce mosaicism.

Other concerns by reviewer #2 related to a lack of comparison of the base editors with other genetic engineering methods, a lack of assessment of health and viability of the embryos, and lack of justification for the choice of multiplexed mutations. The authors responded extensively to each of these points with an appropriate revision of the text. Finally, reviewer #2 was concerned about the verification of the precision of base editing and the rates of error of the genotyping methods

used. The authors reanalyzed their whole genome sequencing (WGS) data and concluded that deaminase-mediated random C-to-T mutations are not present at detectable levels in the FAH mutant cynomolgus macaque fetus. Furthermore, they tested their genotyping in wildtype embryos and were not able to detect any genetic modifications in the target sites, in support of the reliability of their genotyping results. Reviewer #2 (or maybe it was the 3rd reviewer?) provided a number of specific points regarding methods and data presentation to guide the authors in their revision of the manuscript. The authors were able to address each and every one of those points satisfactorily. In result, the revised manuscript is much improved, the data presented is clearer, and all pertinent and important results were thoroughly discussed.

In my opinion, the authors did an excellent job addressing the concerns of the reviewers, and the revised manuscript is significantly improved with respect to the original description. The authors have a track record of innovation and leadership in the development and application of modern, state-of-the-art methods to produce genetically modified non-human primates, and their publications are always highly anticipated and regarded by the scientific community. The proof-of-concept study presented here significantly advances the field in achieving difficult but necessary milestones towards the generation of non-human primate models of polygenic diseases. The results presented clearly demonstrate the feasibility of using CRISPR in combination with precise base editors to manipulate the NHP genome with efficacy, efficiency, and precision. This manuscript will certainly constitute an important contribution to the field and is certain to be highly influential in the field.

Reviewer #2:

Remarks to the Author:

Zhang, Aida, and coauthors sufficiently revised the manuscript "Multiplex precise base editing in cynomolgus monkeys" and adjusted their analysis to warrant a recommendation to be published at Nature Communications. The study is of use to the field and the authors have clarified and improved data analysis and discussion to strengthen the manuscript. Overall, I support publication of the revised manuscript in Nature Communications.

Presentation of editing outcomes were clarified substantially. Specifically, the main test figures are now much more easily interpreted and discussion of sources of error throughout the manuscript were added. Limitations of the study are discussed and provide context to the claims made, which strengthens the manuscript.

The off-target analysis is now more complete within the limits of the time- and resource-intensive experiments. Off target analysis assesses mutations in an unbiased, genome-wide manner, which is appropriate to measure guide-independent base editor off-targets. While the calculation which takes into account the mutational rate across macaque genomes likely overestimates the contribution of SNP accrual that is de novo, the de novo SNPs detected are within range of background mutation accumulation (which is accurately stated by the authors in the main text). The authors would likely need more experiments and comparison to untreated offspring to definitively attribute edits in this range to BE activity (or validation of other calling approaches), but the authors' claims are reasonable and supported by their data and analysis.

While it is unfortunate that the embryos could not be carried further, and more data is not available about the engraftment and development of multiplexed edited embryos, the data provided by the authors is of importance to the field as-is.

The authors discuss that newer, more active editors with increased activity and multiplexed editing capability should be considered to improve multiplexed editing and reduce mosaicism. It is reasonable to not perform more experiments with newer editors, as they are proof-of-concept and still telling of the capability of base editors in the context of editing in the macaque embryo. The

authors also discuss possibility of using other methods (RNP delivery, timing) to increase editing and reduce mosaicism.

Some discussion about the limitations of use of mosaic animal models would be useful. How would the authors expect to use these animals? Would one more generation still need to be generated?

NCOMMS-18-34407A

Detailed response to reviewers' comments

We would like to begin by thanking all of you for taking the time in such a difficult time to re-review our revised manuscript on multiplex precise base editing in cynomolgus monkeys and to provide insightful, encouraging, and constructive comments and suggestions. It's a great honor for us to have such a great evaluation from the experts in this field. As detailed below, we have carefully followed your suggestions and accordingly revised the manuscript. We would like to thank you in advance for once again considering our revised manuscript and hope that our efforts have elevated the work to a point that is suitable for publication.

Reviewer #1 (Remarks to the Author):

Review of: "Multiplex precise base editing in cynomolgus monkeys" (NCOMMS-18-34407A)

Summary: This is a manuscript on the use of simultaneous multiplex base editors to modify up to three target sites across 11 genes/loci in cynomolgus monkey embryos to model complex, multigenic diseases. Results show the high specificity of base editing and demonstrate the feasibility of multiplex base editing for modeling polygenic diseases in primate zygotes. The original manuscript was reviewed by 2 reviewers, who provided extensive comments and suggestions for improvements. The main concerns from Reviewer #1 were related to the preliminary nature of the results, in that no live births of the modified embryos nor phenotypes associated with the mutations were demonstrated. Other concerns were for the lack of rationale and justification for the chosen mutations, and lack of demonstration on how the implanted embryos were screened non-destructively prior to being implanted into surrogate mothers as an essential requirement for the

production of genetically-engineered nonhuman primates. Reviewer #2 was mostly concerned about the lack of clarity in the presentation of the results and the lack of an analysis of off- target mutations.

The authors made a substantial revision to the original submission in response to the comments provided by both reviewers, performing new analyses and addressing all concerns raised. In result, the revised manuscript is much improved. In response to the main concerns by Reviewer #1, the authors argued – successfully in my opinion, that the present study was designed as a proof-of-concept to investigate the feasibility of multiplex introduction of precise CRISPR base editing mutations in embryos of cynomolgus monkey. As it will take several years to produce live monkeys, the demonstration of live births and the associated phenotypic analysis of the induced mutations is by necessity topics for a future study. Not only the proof-of-concept shown here is essential to moving forward with the production of mutant animals, the results already obtained must be shared with the scientific community, specifically to guide other researchers in achieving similar results, and especially to inform the global community of how carefully the authors have been to optimize the methodology as much as possible to ensure that any animal eventually produced will have the intended mutations and to optimize and minimize the number of animals used to produce the invaluable mutant animals. The authors addressed reviewer #1’s concern about embryo transfers well, including detailed information about the embryo transfers and their development relative to embryos that were injected with buffers. In response to the concern by Reviewer #1 that no screening of pre-implantation embryos was presented, the authors answered that only embryos with single mutations were implanted, but no prescreening of such embryos to identify mutations was performed. This is because there are no non- destructive methods to screen NHP embryos available. All genotypic confirmation needs to be performed after birth. As reviewer #1 correctly points out, “the efficiency of double and triple mutations decreases to the point that it is not advisable to transfer the embryos”, and indeed the authors did not transfer any embryos with multiple mutations.

Reviewer #2 was strongly supportive of publication of the manuscript in Nature Communications, stating the manuscript will be an important milestone in the field. Reviewer #2 raised some concerns about the clarity of data presentation, and in particular, the lack of an effort by the authors to identify methods to reduce mosaicism. In response, the authors reorganized data presentation and added discussion about potential ways to reduce mosaicism.

Other concerns by reviewer #2 related to a lack of comparison of the base editors with other genetic engineering methods, a lack of assessment of health and viability of the embryos, and lack of justification for the choice of multiplexed mutations. The authors responded extensively to each of these points with an appropriate revision of the text. Finally, reviewer #2 was concerned about the verification of the precision of base editing and the rates of error of the genotyping methods used. The authors reanalyzed their whole genome sequencing (WGS) data and concluded that deaminase-mediated random C-to-T mutations are not present at detectable levels in the FAH mutant cynomolgus macaque fetus. Furthermore, they tested their genotyping in wildtype embryos and were not able to detect any genetic modifications in the target sites, in support of the reliability of their genotyping results. Reviewer #2 (or maybe it was the 3rd reviewer?) provided a number of specific points regarding methods and data presentation to guide the authors in their revision of the manuscript. The authors were able to address each and every one of those points satisfactorily. In result, the revised manuscript is much improved, the data presented is clearer, and all pertinent and important results were thoroughly discussed.

In my opinion, the authors did an excellent job addressing the concerns of the reviewers, and the revised manuscript is significantly improved with respect to the original description. The authors have a track record of innovation and leadership in the development and application of modern, state-of-the-art methods to produce genetically modified non-human primates, and their publications are always highly anticipated and regarded by the scientific community. The proof-of-concept study presented here significantly advances the field in achieving difficult but necessary

milestones towards the generation of non-human primate models of polygenic diseases. The results presented clearly demonstrate the feasibility of using CRISPR in combination with precise base editors to manipulate the NHP genome with efficacy, efficiency, and precision. This manuscript will certainly constitute an important contribution to the field and is certain to be highly influential in the field.

We very much appreciate this positive evaluation and the acknowledgement of the importance of our work. Thank you for taking the time to rigorously review our manuscript.

Reviewer #2 (Remarks to the Author):

Zhang, Aida, and coauthors sufficiently revised the manuscript “Multiplex precise base editing in cynomolgus monkeys” and adjusted their analysis to warrant a recommendation to be published at Nature Communications. The study is of use to the field and the authors have clarified and improved data analysis and discussion to strengthen the manuscript. Overall, I support publication of the revised manuscript in Nature Communications.

Presentation of editing outcomes were clarified substantially. Specifically, the main test figures are now much more easily interpreted and discussion of sources of error throughout the manuscript were added. Limitations of the study are discussed and provide context to the claims made, which strengthens the manuscript.

The off-target analysis is now more complete within the limits of the time- and resource-intensive experiments. Off target analysis assesses mutations in an unbiased, genome-wide manner, which is appropriate to measure guide-independent base editor off-targets. While the calculation which takes into account the mutational rate across macaque genomes likely overestimates the contribution of SNP accrual that is de novo, the de novo SNPs detected are within range of background mutation accumulation (which is accurately stated by the authors in the main text). The authors would likely need more experiments and comparison to

untreated offspring to definitively attribute edits in this range to BE activity (or validation of other calling approaches), but the authors' claims are reasonable and supported by their data and analysis.

We're sincerely thankful for the very careful and constructive comments that helped us improve our manuscript and we are very pleased to receive such a positive response to our revisions. With regard to addition controls for our off-target WGS analysis, we agree with your suggestion that we should have untreated control monkey or endogenous controls made by 2-cell injection, and will incorporate these into our follow-up studies. Additionally, in this newly revised manuscript we have added a sentence to the discussion emphasize the importance of these controls for off-target analysis.

While it is unfortunate that the embryos could not be carried further, and more data is not available about the engraftment and development of multiplexed edited embryos, the data provided by the authors is of importance to the field as-is.

We sincerely apologize for this. Once COVID-19 is over and we go back to the lab we will resume these projects. We appreciate your understanding of our design of the work as proof-of-concept to investigate the feasibility of multiplex base editing in monkey embryos.

The authors discuss that newer, more active editors with increased activity and multiplexed editing capability should be considered to improve multiplexed editing and reduce mosaicism. It is reasonable to not perform more experiments with newer editors, as they are proof-of-concept and still telling of the capability of base editors in the context of editing in the macaque embryo. The authors also discuss possibility of using other methods (RNP delivery, timing) to increase editing and reduce mosaicism.

Some discussion about the limitations of use of mosaic animal models would be

useful. How would the authors expect to use these animals? Would one more generation still need to be generated?

Thank you very much for suggesting this important point. We believe the mosaic founder animals are still useful for disease modeling and phenotypic analysis. For example, in our recent *SHANK3* knockout cynomolgus monkey study (Yang *et al.*, *Nature* 2019), among 5 founder mutant monkeys, we found 1 non-mosaic homozygous, 1 non-mosaic heterozygous, and 3 mosaics (one had no wildtype allele, thus compound heterozygous, and the other 2 had 40-50% wildtype allele) at genomic DNA level. Western blotting revealed further mosaicism in all founders with reduction of SHANK3 protein levels ranging between 20-90%. Importantly, we observed strong autism-related phenotypes with some variations dependent upon leftover SHANK3 protein in these mosaic founders. In another study of *MECP2* knockout cynomolgus monkeys, all 5 founders were mosaic with mutation frequencies ranged between 30-50% at genomic DNA level and protein reduction ranging between 20-60% (Chen *et al.*, *Cell* 2017). These founder mosaic monkeys also showed clear phenotypes similar to Rett syndrome patients. These studies strongly encourage us to use mosaic founder animals made by base editors for phenotypic analysis, because, in our current study, SaKKH-BE3 and ABE in a multiplexed condition yielded average 80-95% and 50-80% edited allele frequencies (within edited embryos), respectively, that are enough to edit more than half of all alleles throughout body and potentially induce phenotypes. Nonetheless, non-mosaic, cleanly edited founder animals are the end goal and highly efficient base editors such as SaKKH-BE3 combined with earlier, RNP-based delivery should be tested. Also, development of new reproduction technologies such as optimal hormonal priming to accelerate sexual maturation will be critical to produce second-generation non-mosaic monkeys by breeding. We have added this point to the discussion in this second revision of the manuscript.